# Regularization Matters in Policy Optimization - An Empirical Study on Continuous Control

**Zhuang Liu**[1*]**, Xuanlin Li**[1*]**, Bingyi Kang**[2]**, Trevor Darrell**[1]
[1]University of California, Berkeley    [2]National University of Singapore

## Abstract

Deep Reinforcement Learning (Deep RL) has been receiving increasingly more attention thanks to its encouraging performance on a variety of control tasks. Yet, conventional regularization techniques in training neural networks (e.g., $L_2$ regularization, dropout) have been largely ignored in RL methods, possibly because agents are typically trained and evaluated in the same environment, and because the deep RL community focuses more on high-level algorithm designs. In this work, we present the first comprehensive study of regularization techniques with multiple policy optimization algorithms on continuous control tasks. Interestingly, we find conventional regularization techniques on the policy networks can often bring large improvement, especially on harder tasks. We also compare these techniques with the more widely used entropy regularization. Our findings are shown to be robust against training hyperparameter variations. In addition, we study regularizing different components and find that only regularizing the policy network is typically the best. Finally, we discuss and analyze why regularization may help generalization in RL from four perspectives - sample complexity, return distribution, weight norm, and noise robustness. We hope our study provides guidance for future practices in regularizing policy optimization algorithms. Our code is available at https://github.com/xuanlinli17/iclr2021_rlreg.

## 1 Introduction

The use of regularization methods to prevent overfitting is a key technique in successfully training neural networks. Perhaps the most widely recognized regularization methods in deep learning are $L_2$ regularization (also known as weight decay) and dropout (Srivastava et al., 2014). These techniques are standard practices in supervised learning tasks across many domains. Major tasks in computer vision, e.g., image classification (Krizhevsky et al., 2012; He et al., 2016), object detection (Ren et al., 2015; Redmon et al., 2016), use $L_2$ regularization as a default option. In natural language processing, for example, the Transformer (Vaswani et al., 2017) uses dropout, and the popular BERT model (Devlin et al., 2018) uses $L_2$ regularization. In fact, it is rare to see state-of-the-art neural models trained without regularization in a supervised setting.

However, in deep reinforcement learning (deep RL), those conventional regularization methods are largely absent or underutilized in past research, possibly because in most cases we are maximizing the return on the same task as in training. In other words, there is no generalization gap from the training environment to the test environment (Cobbe et al., 2018). Heretofore, researchers in deep RL have focused on high-level algorithm design and largely overlooked issues related to network training, including regularization. For popular policy optimization algorithms like Trust Region Policy Optimization (TRPO) (Schulman et al., 2015), Proximal Policy Optimization (PPO) (Schulman et al., 2017), and Soft Actor Critic (SAC) (Haarnoja et al., 2018), conventional regularization methods were not considered. In popular codebases such as the OpenAI Baseline (Dhariwal et al., 2017), $L_2$ regularization and dropout were not incorporated. Instead, a commonly used regularization in RL is the entropy regularization which penalizes the high-certainty output from the policy network to encourage more exploration and prevent the agent from overfitting to certain actions. The entropy

---

*Equal contribution

regularization was first introduced by (Williams & Peng, 1991) and now used by many contemporary algorithms (Mnih et al., 2016; Schulman et al., 2017; Teh et al., 2017; Farebrother et al., 2018).

In this work, we take an empirical approach to assess the conventional paradigm which omits common regularization when learning deep RL models. We study agent performance on current task (the environment which the agent is trained on), rather than its generalization ability to different environments as in many recent works (Zhao et al., 2019; Farebrother et al., 2018; Cobbe et al., 2018). We specifically focus our study on policy optimization methods, which are becoming increasingly popular and have achieved top performance on various tasks. We evaluate four popular policy optimization algorithms, namely SAC, PPO, TRPO, and the synchronous version of Advantage Actor Critic (A2C), on multiple continuous control tasks. Various conventional regularization techniques are considered, including $L_2/L_1$ weight regularization, dropout, weight clipping (Arjovsky et al., 2017) and Batch Normalization (BN) (Ioffe & Szegedy, 2015). We compare the performance of these regularization techniques to that without regularization, as well as the entropy regularization.

Surprisingly, even though the training and testing environments are the same, we find that many of the conventional regularization techniques, when imposed to the policy networks, can still bring up the performance, sometimes significantly. Among those regularizers, $L_2$ regularization tends to be the most effective overall. $L_1$ regularization and weight clipping can boost performance in many cases. Dropout and Batch Normalization tend to bring improvements only on off-policy algorithms. Additionally, all regularization methods tend to be more effective on more difficult tasks. We also verify our findings with a wide range of training hyperparameters and network sizes, and the result suggests that imposing proper regularization can sometimes save the effort of tuning other training hyperparameters. We further study which part of the policy optimization system should be regularized, and conclude that generally only regularizing the policy network suffices, as imposing regularization on value networks usually does not help. Finally we discuss and analyze possible reasons for some experimental observations. Our main contributions can be summarized as follows:

- To our best knowledge, we provide the first systematic study of common regularization methods in policy optimization, which have been largely ignored in the deep RL literature.
- We find conventional regularizers can be effective on continuous control tasks (especially on harder ones) with statistical significance, under randomly sampled training hyperparameters. Interestingly, simple regularizers ($L_2$, $L_1$, weight clipping) could perform better than entropy regularization, with $L_2$ generally the best. BN and dropout can only help in off-policy algorithms.
- We study which part of the network(s) should be regularized. The key lesson is to regularize the policy network but not the value network.
- We analyze why regularization may help generalization in RL through sample complexity, return distribution, weight norm, and training noise robustness.

## 2 RELATED WORKS

**Regularization in Deep RL.** There have been many prior works studying the theory of regularization in policy optimization (Farahmand et al., 2009; Neu et al., 2017; Zhang et al., 2020). In practice, conventional regularization methods have rarely been applied in deep RL. One rare case of such use is in Deep Deterministic Policy Gradient (DDPG) (Lillicrap et al., 2016), where Batch Normalization is applied to all layers of the actor and some layers of the critic, and $L_2$ regularization is applied to the critic. Some recent studies have developed more complicated regularization approaches to continuous control tasks. Cheng et al. (2019) regularizes the stochastic action distribution $\pi(a|s)$ using a control prior and dynamically adjusts regularization weight based on the temporal difference (TD) error. Parisi et al. (2019) uses TD error regularization to penalize inaccurate value estimation and Generalized Advantage Estimation (GAE) (Schulman et al., 2016) regularization to penalize GAE variance. However, most of these regularizations are rather complicated (Cheng et al., 2019) or catered to certain algorithms (Parisi et al., 2019). Also, these techniques consider regularizing the output of the network, while conventional methods mostly directly regularize the parameters. In this work, we focus on studying these simpler but under-utilized regularization methods.

**Generalization in Deep RL** typically refers to how the model perform in a different environment from the one it is trained on. The generalization gap can come from different modes/levels/difficulties of a game (Farebrother et al., 2018), simulation vs. real world (Tobin et al., 2017), parameter variations (Pattanaik et al., 2018), or different random seeds in environment generation (Zhang et al.,

2018b). There are a number of methods designed to address this issue, e.g., through training the agent over multiple domains/tasks (Tobin et al., 2017; Rajeswaran et al., 2017), adversarial training (Tobin et al., 2017), designing model architectures (Srouji et al., 2018), adaptive training (Duan et al., 2016), etc. Meta RL (Finn et al., 2017; Gupta et al., 2018; Al-Shedivat et al., 2017) try to learn generalizable agents by training on many environments drawn from the same family/distribution. There are also some comprehensive studies on RL generalization with interesting findings (Zhang et al., 2018a;b; Zhao et al., 2019; Packer et al., 2018), e.g., algorithms performing better in training environment could perform worse with domain shift (Zhao et al., 2019).

Recently, several studies have investigated conventional regularization's effect on generalization across tasks. (Farebrother et al., 2018) shows that in Deep Q-Networks (DQN), $L_2$ regularization and dropout are sometime beneficial when evaluated on the same Atari game with mode variations. (Cobbe et al., 2018) shows that $L_2$ regularization, dropout, BN, and data augmentation can improve generalization performance, but to a less extent than entropy regularization and $\epsilon$-greedy exploration. Different from those studies, we focus on regularization's effect in the same environment, yet on which conventional regularizations are under-explored.

## 3 EXPERIMENTS

### 3.1 SETTINGS

**Regularization Methods.** We study six regularization methods, namely, $L_2$ and $L_1$ weight regularization, weight clipping, Dropout (Srivastava et al., 2014), Batch Normalization (Ioffe & Szegedy, 2015), and entropy regularization. See Appendix A for detailed introduction. Note that we consider entropy as a separate regularization method because it encourages exploration and helps to prevent premature convergence (Mnih et al., 2016). In Appendix N, we show that in the presence of certain regularizers, adding entropy on top does not lead to significant performance difference.

**Algorithms.** We evaluate regularization methods on four popular policy optimization algorithms, namely, A2C (Mnih et al., 2016), TRPO (Schulman et al., 2015), PPO (Schulman et al., 2017), and SAC (Haarnoja et al., 2018). The first three algorithms are on-policy while the last one is off-policy. For the first three algorithms, we adopt the code from OpenAI Baseline (Dhariwal et al., 2017), and for SAC, we use the official implementation at (Haarnoja, 2018).

**Tasks.** The algorithms with different regularizers are tested on nine continuous control tasks: Hopper, Walker, HalfCheetah, Ant, Humanoid, and HumanoidStandup from MuJoCo (Todorov et al., 2012); Humanoid, AtlasForwardWalk, and HumanoidFlagrun from RoboSchool (OpenAI). Among the MuJoCo tasks, agents for Hopper, Walker, and HalfCheetah are easier to learn, while Ant, Humanoid, HumanoidStandup are relatively harder (larger state-action space, more training examples). The three Roboschool tasks are even harder than the MuJoCo tasks as they require more timesteps to converge (Klimov & Schulman, 2017). To better understand how different regularization methods work on different difficulties, we roughly categorize the first three environments as "easy" tasks and the last six as "hard" tasks. Besides continuous control, we provide results on randomly sampled Atari environments (Bellemare et al., 2012) in Appendix S, which have discrete action space and different reward properties. Our observations are mostly similar to those on continuous control tasks.

**Training.** On MuJoCo tasks, we keep all hyperparameters unchanged as in the codebase adopted. Since hyperparameters for the RoboSchool tasks are not included, we briefly tune the hyperparameters for each algorithm before we apply regularization (details in Appendix D). For details on regularization strength tuning, please see Appendix C. The results shown in this section are obtained by **only regularizing the policy network**, and a further study on this will be presented in Section 5. We run each experiment independently with five seeds, then use the average return over the last 100 episodes as the final result. Each regularization method is evaluated independently, with other regularizers turned off. We refer to the result without any regularization as the baseline. For BN and dropout, we use its training mode in updating the network, and test mode in sampling trajectories. During training, negligible computation overhead is induced when a regularizer is applied. Specifically, the increase in training time for BN is $\sim 10\%$, dropout $\sim 5\%$, while $L_2$, $L_1$, weight clipping, and entropy regularization are all $< 1\%$. We used up to 16 NVIDIA Titan Xp GPUs and 96 Intel Xeon E5-2667 CPUs, and all experiments take roughly 57 days with resources fully utilized.

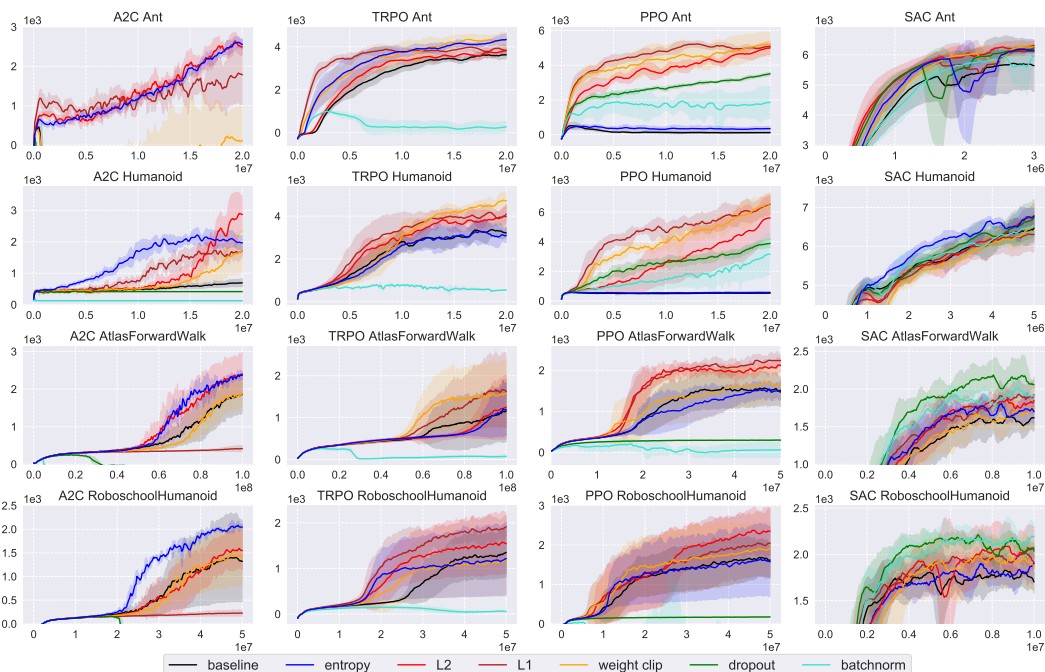

Figure 1: Return vs. timesteps, for four algorithms (columns) and four environments (rows).

**Additional Notes. 1.** Note that entropy regularization is still applicable for SAC, despite it already incorporates the maximization of entropy in the reward. In our experiments, we add the entropy regularization term to the policy loss function in equation (12) of (Haarnoja et al., 2018). **2.** In our experiments, $L_2$ regularization loss is added to the training loss, which is then optimized using Adam (Kingma & Ba, 2015). (Loshchilov & Hutter, 2019) observes that $L_2$ regularization interacts poorly with Adam and proposes AdamW to decouple weight decay from the optimization steps. However, in policy optimization algorithms, we find that the performance of AdamW with decoupled weight decay is slightly worse than the performance of Adam with $L_2$ loss directly added. Comparisons are shown in Appendix O. **3.** Policy network dropout is not applicable to TRPO because during policy updates, different neurons in the old and new policy networks are dropped out, causing different shifts in the old and new action distributions given the same state, which violates the trust region constraint. In this case, the algorithm fails to perform any update from network initialization.

## 3.2   RESULTS

**Training curves.** We plot the training curves from four environments (rows) in Figure 1, on four algorithms (columns). Figures for the rest five environments are deferred to Appendix P. In the figure, different colors are used to denote different regularization methods, e.g., black is the baseline method. Shades are used to denote $\pm 1$ standard deviation range. Notably, these conventional regularizers can frequently boost the performance across different tasks and algorithms, demonstrating that a study on the regularization in deep RL is highly demanding. We observe that BN always significantly hurts the baseline for on-policy algorithms. The reason will be discussed later. For the off-policy SAC algorithm, dropout and BN sometimes bring large improvement on hard tasks like AtlasForwardWalk and RoboschoolHumanoid. Interestingly, in some cases where the baseline (with the default hyperparameters in the codebase) does not converge to a reasonable solution, e.g., A2C Ant, PPO Humanoid, imposing some regularization can make the training converge to a high level.

**How often do regularizations help?** To quantitatively measure the effectiveness of the regularizations on each algorithm across different tasks, we **define the condition when a regularization is said to "improve" upon the baseline** in a certain environment. Denote the baseline mean return over five seeds on an environment as $\mu_{\text{env},b}$, and the mean and standard deviation of the return obtained with a certain regularization method over five seeds as $\mu_{\text{env},r}$ and $\sigma_{\text{env},r}$. We say the performance is "improved" by the regularization if $\mu_{\text{env},r} - \sigma_{\text{env},r} > \max(\mu_{\text{env},b}, T(\text{env}))$, where $T(\text{env})$ is the

minimum return threshold of an environment. The threshold serves to ensure the return is at least in a reasonable level. We set the threshold to be $10^5$ for HumanoidStandup and $10^3$ for all other tasks.

Table 1: Percentage (%) of environments where the final performance "improves" with regularization, by our definition in Section 3.2.

| Reg \ Alg | A2C | | | TRPO | | | PPO | | | SAC | | | TOTAL | | |
|---|---|---|---|---|---|---|---|---|---|---|---|---|---|---|---|
| | Easy | Hard | Total | Easy | Hard | Total | Easy | Hard | Total | Easy | Hard | Total | Easy | Hard | Total |
| Entropy | **33.3** | **100.0** | **77.8** | 0.0 | 50.0 | 33.3 | 0.0 | 33.3 | 22.2 | 33.3 | 50.0 | 44.4 | 16.7 | 58.3 | 44.4 |
| $L_2$ | 0.0 | 50.0 | 33.3 | 0.0 | **66.7** | **44.4** | **33.3** | **83.3** | **66.7** | **66.7** | **66.7** | **66.7** | 25.0 | **66.7** | **52.8** |
| $L_1$ | 0.0 | 50.0 | 33.3 | 0.0 | **66.7** | **44.4** | **33.3** | 66.7 | 55.6 | 33.3 | 50.0 | 44.4 | 16.7 | 58.3 | 44.4 |
| Weight Clip | 0.0 | 16.7 | 11.1 | **33.3** | 33.3 | 33.3 | **33.3** | 66.7 | 55.6 | 33.3 | 16.7 | 22.2 | 25.0 | 33.3 | 30.6 |
| Dropout | 0.0 | 0.0 | 0.0 | N/A | N/A | N/A | **33.3** | 50.0 | 44.4 | **66.7** | 50.0 | 55.6 | **33.3** | 33.3 | 33.3 |
| BatchNorm | 0.0 | 0.0 | 0.0 | 0.0 | 0.0 | 0.0 | 0.0 | 16.7 | 11.1 | 33.3 | 50.0 | 44.4 | 8.3 | 16.7 | 13.9 |

The results are shown in Table 1. Perhaps the most significant observation is that $L_2$ regularization is the most often to improve upon the baseline. A2C algorithm is an exception, where entropy regularization is the most effective. $L_1$ regularization behaves similar to $L_2$ regularization, but is outperformed by the latter. Weight clipping's usefulness is highly dependent on the algorithms and environments. Despite in total it only helps at 30.6% times, it can sometimes outperform entropy regularization by a large margin, e.g., in TRPO Humanoid and PPO Humanoid as shown in Figure 1. BN is not useful at all in the three on-policy algorithms (A2C, TRPO, and PPO). Dropout is not useful in A2C at all, and sometimes helps in PPO. However, BN and dropout can be useful in SAC. All regularization methods generally improve more often when they are used on harder tasks, perhaps because for easier ones the baseline is often sufficiently strong to reach a high performance.

Note that under our definition, not "improving" does not indicate "hurting". If we define "hurting" as $\mu_{\text{env},r} + \sigma_{\text{env},r} < \mu_{\text{env},b}$ (the return minimum threshold is not considered here), then total percentage of hurting is 0.0% for $L_2$, 2.8% for $L_1$, 5.6% for weight clipping, 44.4% for dropout, 66.7% for BN, and 0.0% for entropy. In other words, under our parameter tuning range, $L_2$ and entropy regularization never hurt with appropriate strengths. For BN and dropout, we also note that almost all hurting cases are in on-policy algorithms, except one case for BN in SAC. In sum, all regularizations in our study very rarely hurt the performance except for BN/dropout in on-policy methods.

**How much do regularizations improve?** For each algorithm and environment (for example, PPO on Ant), we calculate a $z$-score for each regularization method and the baseline, by treating results produced by all regularizations (including baseline) and all five seeds together as a population, and calculate each method's average $z$-scores from its five final results (positively clipped). $z$-score is also known as "standard score", the signed fractional number of standard deviations by which the value of a data point is above the mean value. For each algorithm and environment, a regularizer's $z$-score roughly measures its relative performance among others. The $z$-scores are then averaged over environments of a certain difficulty (easy/hard), and the results are shown in Table 2. In terms of the average improved margin, we can draw mostly similar observations as the improvement frequency (Table 1): $L_2$ tops the average $z$-score most often, and by large margin in total; entropy regularization is best used with A2C; Dropout and BN are only useful in the off-policy SAC algorithm; the improvement over baseline is larger on hard tasks. Notably, for all algorithms, any regularization on average outperforms the baseline on hard tasks, except dropout and BN in on-policy algorithms. On hard tasks, $L_1$ and weight clipping also perform higher than entropy in total, besides $L_2$. To

Table 2: Average $z$-scores. Note that a negative $z$-score does not necessarily mean the method hurts, because it could be higher than the baseline. The scores within 0.01 range from the highest are in **bold**.

| Reg \Alg | A2C | | | TRPO | | | PPO | | | SAC | | | TOTAL | | |
|---|---|---|---|---|---|---|---|---|---|---|---|---|---|---|---|
| | Easy | Hard | Total | Easy | Hard | Total | Easy | Hard | Total | Easy | Hard | Total | Easy | Hard | Total |
| Baseline | 0.30 | -0.17 | -0.02 | 0.28 | 0.10 | 0.16 | 0.24 | -0.54 | -0.28 | -0.22 | -0.47 | -0.39 | 0.15 | -0.27 | -0.13 |
| Entropy | **1.14** | **1.01** | **1.06** | 0.16 | 0.30 | 0.26 | **0.43** | -0.25 | -0.02 | 0.32 | -0.16 | 0.00 | **0.51** | 0.23 | 0.32 |
| $L_2$ | 0.53 | 0.93 | 0.80 | **0.51** | 0.39 | 0.43 | 0.30 | **0.76** | **0.61** | **0.36** | 0.25 | 0.28 | 0.43 | **0.58** | **0.53** |
| $L_1$ | 0.15 | 0.43 | 0.34 | 0.31 | **0.57** | **0.48** | 0.27 | **0.76** | **0.60** | 0.19 | -0.17 | -0.05 | 0.23 | 0.40 | 0.34 |
| Weight Clip | 0.22 | 0.24 | 0.24 | 0.28 | 0.49 | 0.42 | 0.34 | 0.63 | 0.53 | -0.36 | -0.09 | -0.18 | 0.12 | 0.32 | 0.25 |
| Dropout | -1.16 | -1.18 | -1.17 | N/A | N/A | N/A | -0.12 | -0.47 | -0.35 | 0.35 | **0.49** | **0.44** | -0.31 | -0.39 | -0.36 |
| BatchNorm | -1.19 | -1.26 | -1.24 | -1.54 | -1.85 | -1.75 | -1.47 | -0.89 | -1.08 | -0.64 | 0.17 | -0.10 | -1.21 | -0.96 | -1.04 |

further verify our observations, we present $z$-scores for MuJoCo environments in Appendix G where we increase the number of seeds from 5 to 10. Our observations are consistent with those in Table 2.

Besides the improvement percentage (Table 1) and the $z$-score (Table 2), we provide more metrics of comparison (e.g., average ranking, min-max scaled return) to comprehensively compare the different regularization methods. We also conduct **statistical significance** tests on these metrics, and the improvement are mostly statistically significant ($p < 0.05$). We believe evaluating under a variety of metrics make our conclusions more reliable. Detailed results are in Appendix F, I, and J. In addition, we provide detailed justification in Appendix K that, because we test on the entire set of environments instead of on a single environment, our sample size is large enough to satisfy the condition of significance tests and provide reliable results.

# 4 ROBUSTNESS WITH HYPERPARAMETER CHANGES

In the previous section, the experiments are conducted mostly with the default hyperparameters in the codebase we adopt, which are not necessarily optimized. For example, PPO Humanoid baseline performs poorly using default hyperparameters, not converging to a reasonable solution. Meanwhile, it is known that RL algorithms are very sensitive to hyperparameter changes (Henderson et al., 2018). Thus, our findings can be vulnerable to such variations. To further confirm our findings, we evaluate the regularizations under a variety of hyperparameter settings. For each algorithm, we sample five hyperparameter settings for the baseline and apply regularization on each of them. Due to the heavy computation cost, we only evaluate on five environments: Hopper, Walker, Ant, Humanoid, HumanoidStandup. Under our sampled hyperparameters, poor baselines are mostly significantly improved. See Appendix E/ Q for details on sampling and curves. The $z$-scores are shown in Table 3. We note that our main findings in Section 3 still hold. Interestingly, compared to the previous section, $L_2$, $L_1$, and weight clipping all tend to be better than entropy regularization by larger margins. For the $p$-scores of statistical significance/improvement percentages, see Appendix F/H.

Table 3: The average $z$-score for each regularization method, under five sampled hyperparameter settings.

| Reg \Alg | A2C | | | TRPO | | | PPO | | | SAC | | | TOTAL | | |
|---|---|---|---|---|---|---|---|---|---|---|---|---|---|---|---|
| | Easy | Hard | Total | Easy | Hard | Total | Easy | Hard | Total | Easy | Hard | Total | Easy | Hard | Total |
| Baseline | 0.49 | -0.05 | 0.17 | 0.15 | 0.14 | 0.14 | 0.34 | -0.27 | -0.03 | -0.01 | -0.25 | -0.15 | 0.24 | -0.11 | 0.03 |
| Entropy | 0.42 | 0.52 | 0.48 | 0.19 | 0.26 | 0.24 | 0.14 | -0.14 | -0.03 | **0.21** | -0.12 | 0.01 | 0.24 | 0.13 | 0.17 |
| $L_2$ | 0.08 | **0.82** | 0.52 | 0.36 | 0.48 | **0.43** | 0.52 | 0.86 | 0.72 | 0.02 | **0.27** | **0.17** | 0.24 | **0.61** | **0.46** |
| $L_1$ | **0.53** | 0.71 | **0.64** | 0.24 | **0.51** | 0.41 | 0.44 | 0.77 | 0.64 | 0.12 | 0.07 | 0.09 | **0.33** | 0.51 | 0.44 |
| Weight Clip | 0.45 | 0.50 | 0.48 | **0.49** | 0.41 | **0.44** | 0.23 | 0.52 | 0.40 | -0.50 | -0.00 | -0.20 | 0.17 | 0.36 | 0.28 |
| Dropout | -0.24 | -1.07 | -0.74 | N/A | N/A | N/A | -0.92 | -0.83 | -0.87 | 0.01 | -0.10 | -0.06 | -0.38 | -0.67 | -0.55 |
| BatchNorm | -1.74 | -1.42 | -1.54 | -1.43 | -1.81 | -1.66 | -0.75 | -0.91 | -0.85 | 0.16 | 0.14 | 0.15 | -0.94 | -1.00 | -0.98 |

To better visualize the robustness against change of hyperparameters, we show the result when a single hyperparameter is varied in Figure 2. We note that the certain regularizations can consistently improve the baseline with different hyperparameters. In these cases, proper regularizations can ease the hyperparameter tuning process, as they bring up performance of baselines with suboptimal hyperparameters to be higher than that with better ones.

# 5 POLICY AND VALUE NETWORK REGULARIZATION

Table 4: Percentage (%) of environments where performance "improves" when regularized on policy / value / policy and value networks.

| Reg\Alg | A2C | | | TRPO | | | PPO | | | SAC | | | TOTAL | | |
|---|---|---|---|---|---|---|---|---|---|---|---|---|---|---|---|
| | Pol | Val | P+V | Pol | Val | P+V | Pol | Val | P+V | Pol | Val | P+V | Pol | Val | P+V |
| $L_2$ | **50.0** | 0.0 | 16.7 | **50.0** | 16.7 | 33.3 | **66.7** | 16.7 | **66.7** | **66.7** | 33.3 | 33.3 | **58.3** | 16.7 | 37.5 |
| $L_1$ | **50.0** | 16.7 | **50.0** | 33.3 | 0.0 | **33.3** | **66.7** | 0.0 | 50.0 | 33.3 | 33.3 | 33.3 | 45.8 | 12.5 | 41.7 |
| Weight Clip | 16.7 | 0.0 | 16.7 | **50.0** | 33.3 | 16.7 | **66.7** | 0.0 | **66.7** | 33.3 | 16.7 | 16.7 | 41.7 | 8.3 | 29.2 |
| Dropout | 0.0 | **16.7** | 0.0 | N/A | **33.3** | N/A | **66.7** | 33.3 | 50.0 | **50.0** | 0.0 | 0.0 | 38.9 | 20.8 | 16.7 |
| BatchNorm | 16.7 | **16.7** | **16.7** | 0.0 | **16.7** | 0.0 | 16.7 | 0.0 | 50.0 | 33.3 | 16.7 | 0.0 | 16.7 | 12.5 | **16.7** |

Our experiments so far only impose regularization on policy network. To investigate the relationship between policy and value network regularization, we compare four options: 1) no regularization, and regularizing 2) policy network, 3) value network, 4) policy and value networks. For 2) and 3) we tune

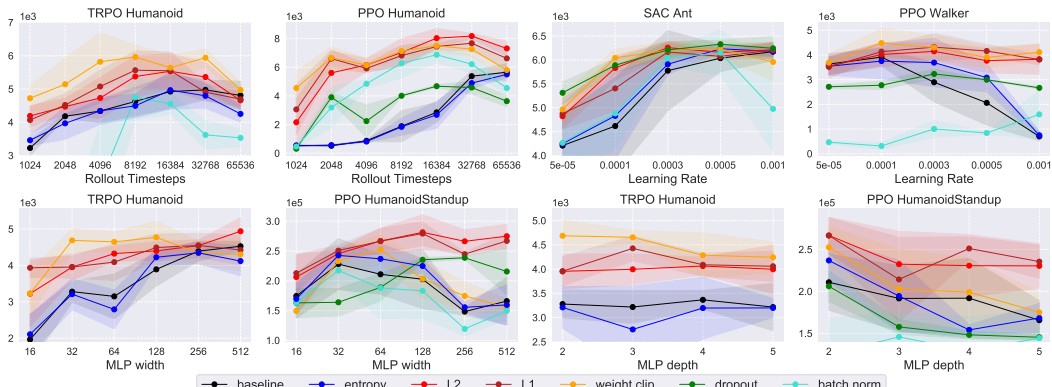

Figure 2: Final return vs. single hyperparameter change. "Rollout Timesteps" refers to the number of state-action samples used for training between policy updates.

the regularization strengths independently and then use the appropriate ones for 4) (more details in Appendix C). We evaluate all four algorithms on the six MuJoCo tasks and present the improvement percentage in Table 4. Note that entropy regularization is not applicable to the value network. We observe that generally, only regularizing the policy network is the most often to improve almost all algorithms and regularizations. Regularizing the value network alone does not bring improvement as often as other options. Though regularizing both is better than regularizing value network alone, it is worse than only regularizing the policy network. For detailed training curves, refer to Appendix R.

We also note that the policy optimization algorithms in our study have adopted multiple techniques to train the value function. For example, SAC uses the replay buffer and the clipped double-Q learning. A2C, TRPO, and PPO adopt multi-step roll-out, and the sum of discounted rewards is used as the value network objective. However, analyzing the individual effects of these techniques is not the main focus of our current work. We would like to leave the interaction between these techniques and value network regularization for future work.

## 6 ANALYSIS AND CONCLUSION

**Why does regularization benefit policy optimization?** In RL, when we are training and evaluating on the same environment, there is no generalization gap across different environments. However, there is still generalization between samples: the agents is only trained on the limited trajectories it has experienced, which cannot cover the whole state-action space of the environment. A successful policy needs to generalize from seen samples to unseen ones, which potentially makes regularization necessary. This might also explain why regularization could be more helpful on harder tasks, which have larger state space, and the portion of the space that have appeared in training tends to be smaller. We study how regularization helps generalization through the following perspectives:

**Sampling Complexity.** We compare the return with varying number of training samples/timesteps, since the performance of learning from fewer samples is closely related to generalization ability. From the results in Figure 3, we find that for regularized models to reach the same return level as baseline, they need much fewer training samples. This suggests that certain regularizers can significantly reduce the sampling complexity of baseline and thus lead to better generalization.

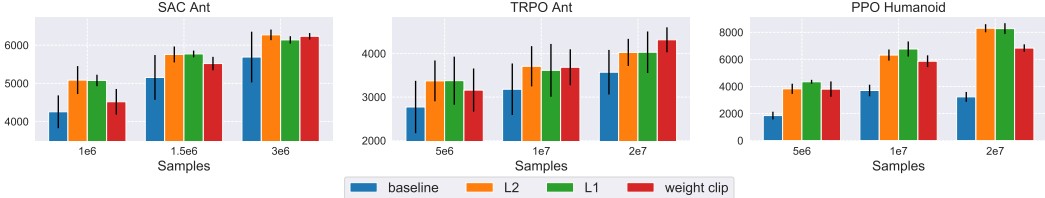

Figure 3: Return with different amount of training samples with error bars from 10 random seeds. Regularized models can reach similar performance as baseline with less data, showing their stronger generalization ability.

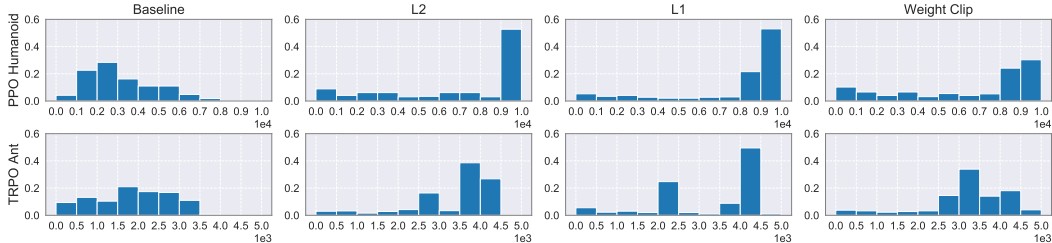

Figure 4: Return distribution (frequency vs. return value) over 100 trajectories. Regularized models generalize to unseen samples more stably with high return.

**Return Distribution.** We evaluate agents trained with and without regularization on 100 different trajectories and plot the return distributions over trajectories in Figure 4. These trajectories represent unseen samples during training, since the state space is continuous (so it is impossible to traverse identical trajectories). For baseline, some trajectories yield relatively high returns, while others yield low returns, demonstrating the baseline cannot stably generalize to unseen examples; for regularized models, the returns are more concentrated at a high level, demonstrating they can more stably generalize to unseen samples. This suggests that certain conventional regularizers can improve the model's generalization ability to larger portion of unseen samples.

**Weight Norm.** We observe that on many tasks, smaller policy weight norm correlates with better generalization ability. An example is illustrated in Table 5 and Figure 5. We observe that $L_2$ regularization accomplishes the effect of entropy regularization and, at the same time, limits the policy norm. Even though both the entropy-regularized model and the $L_2$-regularized model have similar final policy entropy, $L_2$-regularized model have much higher final performance, which suggests that simply increasing the policy entropy is not enough. We conjecture that $L_2$-encouraged small weight norm makes the network less prone to overfitting and provides a better optimization landscape for the model.

Table 5: Comparison of final performance, policy entropy, and policy weight norm on PPO Humanoid.

| Reg | Return | Entropy | Policy Norm |
|---|---|---|---|
| Baseline | 3485±302 | -10.32 | 30.73 |
| Entropy | 3805±349 | 4.46 | 30.97 |
| $L_2$ | 8148±335 | 8.11 | 8.71 |

Table 6: Effect of data augmentation on final performance on PPO Humanoid.

| | Baseline | $L_2$ |
|---|---|---|
| w/o DA | 3485±302 | 8148±335 |
| w/ DA | 3483±293 | 9006±145 |

**Robustness to Training Noise.** Recent works (Kostrikov et al., 2020; Laskin et al., 2020) have applied data augmentation (DA) to RL, mainly on image-based inputs, to improve data efficiency and generalization. Laskin et al. (2020) adds noise to state-based input observations by random scaling them as a form of DA. We apply this technique to both baseline and $L_2$ regularization on PPO Humanoid. At each time step, we randomly scale the input state by a factor of $s$, where $s \sim \text{Unif}(1-k, 1+k)$, $k \in \{0.05, 0.1, 0.2, 0.4, 0.6, 0.8\}$. We select the $k$ with the highest performance on the original environment and report the results in Table 6. Interestingly, while DA cannot improve the baseline performance, it can significantly improve the performance of $L_2$-regularized model. This suggests $L_2$ regularizer can make the model robust to, or even benefit from, noisy/augmented input during training.

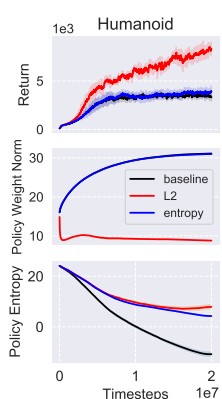

Figure 5: Return, policy network $L_2$ norm, and policy entropy for PPO Humanoid.

**Why do BN and dropout work only with off-policy algorithms?** One finding in our experiments is BN and dropout can sometimes improve on the off-policy algorithm SAC, but mostly hurt on-policy algorithms. We further confirm this observation through experiments on Deep Deterministic Policy Gradient (DDPG, Lillicrap et al. (2016)), another off-policy algorithm, and present the results in Appendix M. We hypothesize two possible reasons: 1) for both BN and dropout, training mode is used to train the network, and testing mode is used to sample actions during interaction with the environment, leading to a discrepancy between the sampling policy and optimization policy (the same holds if we always use training mode). For on-policy algorithms, if such discrepancy is large, it can cause severe "off-policy issues", which hurts the optimization process or even crashes it since their theory necessitates that the data

is "on policy", i.e., data sampling and optimization policies are the same. For off-policy algorithms, this discrepancy is not an issue, since they sample data from replay buffer and do not require the two policies to be the same. 2) BN can be sensitive to input distribution shifts, since the mean and std statistics depend on the input, and if the input distribution changes too quickly in training, the mapping functions of BN layers can change quickly too, which can possibly destabilize training. One evidence for this is that in supervised learning, when transferring a ImageNet pretrained model to other vision datasets, sometimes the BN layers are fixed (Yang et al., 2017) and only other layers are trained. In off-policy algorithms, the sample distributions are relatively slow-changing since we always draw from the whole replay buffer which holds cumulative data; in on-policy algorithms, we always use the samples generated from the latest policy, and the faster-changing input distribution for on-policy algorithms could be harmful to BN.

**In summary**, we conducted the first systematic study of regularization methods on multiple policy optimization algorithms. We found that conventional regularizations ($L_2$, $L_1$, weight clipping) could be effective at improving performance, sometimes more than entropy regularization. BN and dropout could be useful but only on off-policy algorithms. Our findings were confirmed with multiple sampled hyperparameters. Further experiments have shown that generally, the best practice is to regularize the policy network but not the value network or both. Finally we analyze why regularization can help in RL with experiments and discussions.

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

# APPENDIX

## TABLE OF CONTENTS

## A   REGULARIZATION METHODS

There are in general two types of common approaches for imposing regularization. One is to discourage complex models (e.g., weight regularization, weight clipping), and the other is to inject certain noise in network activations (e.g., dropout and Batch Normalization). Here we briefly introduce the methods we investigate in our experiments.

$L_2$ / $L_1$ **Weight Regularization.** Suppose $L$ is the original empirical loss we want to minimize. When applying $L_2$ regularization, we add an additional $L_2$-norm squared loss term $\frac{1}{2}\lambda||\theta||_2^2$ to $L$, where $\theta$ are model parameters and $\lambda$ is a hyperparameter. Similarly, in the case of $L_1$ weight regularization, the additional loss term is $\lambda||\theta||_1$. In our experiments, the total loss is optimized using Adam (Kingma & Ba, 2015). Using $L_2/L_1$ regularization can encourage the model to be simpler/sparse. From a Bayesian view, they impose certain prior distributions on model weights.

**Weight Clipping.** Weight clipping is a simple operation: after each gradient update step, each individual weight is clipped to range $[-c, c]$, where $c$ is a hyperparameter. This could be formally described as $\theta_i \leftarrow \max(\min(\theta_i, c), -c)$. In Wasserstein GANs (Arjovsky et al., 2017), weight clipping is used to enforce the constraint of Lipschitz continuity. This plays an important role in stabilizing the training of GANs (Goodfellow et al., 2014) Weight clipping can also be seen as a regularizor since it reduce the complexity of the model space, by preventing any weight's magnitude from being larger than $c$.

**Dropout.** Dropout (Srivastava et al., 2014) is one of the most successful regularization techniques developed specifically for neural networks. During training, a certain percentage of neurons is deactivated; during testing, all neurons in the neural network are kept, and rescaling is applied to ensure the scale of the activations is the same as training. One explanation for its effectiveness in reducing overfitting is they can prevent "co-adaptation" of neurons. In the policy optimization algorithms we investigate, when the policy or the value network performs updates using minibatches of trajectory data or replay buffer data, we use the training mode of dropout. When the policy network samples trajectories from the environment, we use the testing mode of dropout.

**Batch Normalization.** Batch Normalization (BN) (Ioffe & Szegedy, 2015) is invented to address the problem of "internal covariate shift", and it does the following transformation: $\hat{z} = \frac{z_{in} - \mu_\mathcal{B}}{\sqrt{\sigma_\mathcal{B}^2 + \epsilon}}$; $z_{out} = \gamma\hat{z} + \beta$, where $\mu_\mathcal{B}$ and $\sigma_\mathcal{B}$ are the mean and standard deviation of input activations over $\mathcal{B}$, and $\gamma$ and $\beta$ are trainable affine transformation parameters. BN turns out to greatly accelerate the convergence and bring up the accuracy. It also acts as a regularizer (Ioffe & Szegedy, 2015): during training, the statistics $\mu_\mathcal{B}$ and $\sigma_\mathcal{B}$ depend on the current batch, and BN subtracts and divides different values in each iteration. This stochasticity can encourage subsequent layers to be robust against such input variation. In policy optimization algorithms, we switch between training and testing modes the same way as we do in dropout.

**Entropy Regularization.** In a policy optimization framework, the policy network is used to model a conditional distribution over actions, and entropy regularization is widely used to prevent the learned policy from overfitting to one or some of the actions. More specifically, in each step, the output distribution of the policy network is penalized to have a high entropy. Policy entropy is calculated at each step as $H_{s_i} = -\mathbb{E}_{a_i \sim \pi(a_i|s_i)} \log \pi(a_i|s_i)$, where $(s_i, a_i)$ is the state-action pair. Then the per-sample entropy is averaged within the batch of state-action pairs to obtain the regularization term $L^H = \frac{1}{N} \sum_{s_i} H_{s_i}$. A coefficient $\lambda$ is also needed, and $\lambda L^H$ is added to the policy objective $J(\theta)$. The sum is then maximized during policy updates. Entropy regularization also encourages exploration and prevents premature convergence due to increased randomness in actions, leading to better performance in the long run.

## B  POLICY OPTIMIZATION ALGORITHMS

The policy optimization family of algorithms is one of the most popular methods for solving reinforcement learning problems. It directly parameterizes and optimizes the policy to gain more cumulative rewards. Below, we give a brief introduction to the algorithms we evaluate in our work.

**A2C.**  Sutton et al. (2000) developed a policy gradient to update the parametric policy in a gradient descent manner. However, the gradient estimated in this way suffers from high variance. Advantage Actor Critic (A3C) (Mnih et al., 2016) is proposed to alleviate this problem by introducing a function approximator for values and replacing the Q-values with advantage values. A3C also utilizes multiple actors to parallelize training. The only difference between A2C and A3C is that in a single training iteration, A2C waits for parallel actors to finish sampling trajectories before updating the neural network parameters, while A3C updates in an asynchronous manner.

**TRPO.**  Trust Region Policy Optimization (TRPO) (Schulman et al., 2015) proposes to constrain each update within a safe region defined by KL divergence to guarantee policy improvement during training. Though TRPO is promising at obtaining reliable performance, approximating the KL constraint is quite computationally heavy.

**PPO.**  Proximal Policy Optimization (PPO) (Schulman et al., 2017) simplifies TRPO and improves computational efficiency by developing a surrogate objective that involves clipping the probability ratio to a reliable region, so that the objective can be optimized using first-order methods.

**SAC.**  Soft Actor Critic (SAC) (Haarnoja et al., 2018) optimizes the maximum entropy objective in reward (Ziebart et al., 2008), which is different from the objective of the on-policy methods above. SAC combines soft policy iteration, which maximizes the maximum entropy objective, and clipped double $Q$ learning (Fujimoto et al., 2018), which prevents overestimation bias, during actor and critic updates, respectively.

## C    REGULARIZATION IMPLEMENTATION & TUNING DETAILS

As mentioned in the paper, in Section 3 we only regularize the policy network; in Section 5, we investigate regularizing both policy and value networks.

For $L_1$ and $L_2$ regularization, we add $\lambda \|\theta\|_1$ and $\frac{\lambda}{2}\|\theta\|_2^2$, respectively, to the loss of policy network or value network of each algorithm (for SAC's value regularization, we apply regularization only to the $V$ network instead of also to the two $Q$ networks). $L_1$ and $L_2$ loss are applied to all the weights of the policy or value network. For A2C, TRPO, and PPO, we tune $\lambda$ in the range of $[1e-5, 5e-5, 1e-4, 5e-4]$ for $L_1$ and $[5e-5, 1e-4, 5e-4, 1e-3]$ for $L_2$. For SAC, we tune $\lambda$ in the range of $[5e-4, 1e-3, 5e-3, 1e-2]$ for $L_1$ and $[1e-3, 5e-3, 1e-2, 5e-2]$ for $L_2$.

For weight clipping, the OpenAI Baseline implementation of the policy network of A2C, TRPO, and PPO outputs the mean of policy action from a two-layer fully connected network (MLP). The log standard deviation of the policy action is represented by a standalone trainable vector. We find that when applied only to the weights of MLP, weight clipping makes the performance much better than when applied to only the logstd vector or both. Thus, for these three algorithms, the policy network weight clipping results shown in all the sections above come from clipping only the MLP part of the policy network. On the other hand, in the SAC implementation, both the mean and the log standard deviation come from the same MLP, and there is no standalone log standard deviation vector. Thus, we apply weight clipping to all the weights of the MLP. For all algorithms, we tune the policy network clipping range in $[0.1, 0.2, 0.3, 0.5]$. For value network, the MLP produces a single output of estimated value given a state, so we clip all the weights of the MLP. For A2C, TRPO, and PPO, we tune the clipping range in $[0.1, 0.2, 0.3, 0.5]$. For SAC, we only clip the $V$ network and do not clip the two $Q$ networks for simplicity. We tune the clipping range in $[0.3, 0.5, 0.8, 1.0]$ due to its weights having larger magnitude.

For BatchNorm/dropout, we apply it before the activation function of each hidden layer/immediately after the activation function. When the policy or the value network is performing update using minibatches of trajectory data or minibatches of replay buffer data, we use the train mode of regularization and update the running mean and standard deviation. When the policy is sampling trajectory from the environment, we use the test mode of regularization and use the existing running mean and standard deviation to normalize data. For Batch Normalization/dropout on value network, only training mode is applied since value network does not participate in sampling trajectories. Note that adding policy network dropout on TRPO causes the KL divergence constraint $\mathbb{E}_{s \sim \rho_{\theta_{\text{old}}}}\left[D_{\text{KL}}\left(\pi_{\theta_{\text{old}}}(\cdot|s)\|\pi_\theta(\cdot|s)\right)\right] \leq \delta$ to be violated almost every time during policy network update. Thus, policy network dropout causes the training to fail on TRPO, as the policy network cannot be updated.

For entropy regularization, we add $-\lambda L^H$ to the policy loss. $\lambda$ is tuned from $[5e-5, 1e-4, 5e-4, 1e-3]$ for A2C, TRPO, PPO and $[0.1, 0.5, 1.0, 5.0]$ for SAC. Note that for SAC, our entropy regularization is added directly on the optimization objective (equation 12 in Haarnoja et al. (2018)), and is different from the original maximum entropy objective inside the reward term.

Note that for the three on-policy algorithms (A2C, TRPO, PPO) we use the same tuning range, and the only exception is the off-policy SAC. The reason why SAC's tuning range is different is that SAC uses a hyperparameter that controls the scaling of the reward signal, while A2C, TRPO, and PPO do not. In the original implementation of SAC, the reward signals are pre-tuned to be scaled up by a factor ranging from 5 to 100, according to specific environments. Also, unlike A2C, TRPO, and PPO, SAC uses unnormalized reward because if the reward magnitude is small, then, according to the original paper, the policy becomes almost uniform. Due to the above reasons, the reward magnitude of SAC is much higher than the magnitude of rewards used by A2C, TRPO, and PPO. Thus, the policy network loss and the value network loss have larger magnitude than those of A2C, TRPO, and PPO, so the appropriate regularization strengths become higher. Considering the SAC's much larger reward magnitude, we selected a different range of hyperparameters for SAC before we ran the whole experiments.

The optimal policy network regularization strength we selected for each algorithm and environment used in Section 3 can be seen in the legends of Appendix R. In addition to the results with environment-specific strengths presented in Section 3, we also present the results when the regularization strength is fixed across all environments for the same algorithm. The results are shown in Appendix L.

In Section 5, to investigate the effect of regularizing both policy and value networks, we combine the tuned optimal policy and value network regularization strengths. The detailed training curves are presented in Appendix R.

As a side note, when training A2C, TRPO, and PPO on the HalfCheetah environment, the results have very large variance. Thus, for each regularization method, after we obtain the best strength, we rerun it for another five seeds as the final result in Table 1 and 2.

## D  DEFAULT HYPERPARAMETER SETTINGS

**Training timesteps.** For A2C, TRPO, and PPO, we run $5e6$ timesteps for Hopper, Walker, and HalfCheetah; $2e7$ timesteps for Ant, Humanoid (MuJoCo), and HumanoidStandup; $5e7$ timesteps for Humanoid (RoboSchool); and $1e8$ timesteps for AtlasForwardWalk and HumanoidFlagrun. For SAC, since its simulation speed is much slower than A2C, TRPO, and PPO (as SAC updates its policy and value networks using a minibatch of replay buffer data at every timestep), and since it takes fewer timesteps to converge, we run $1e6$ timesteps for Hopper and Walker; $3e6$ timesteps for HalfCheetah and Ant; $5e6$ timesteps for Humanoid and HumanoidStandup; and $1e7$ timesteps for the RoboSchool environments.

**Hyperparameters for RoboSchool.** In the original PPO paper (Schulman et al., 2017), hyperparameters for the Roboschool tasks are given, so we apply the same hyperparameters to our training, except that instead of linear annealing the log standard deviation of action distribution from $-0.7$ to $-1.6$, we let it to be learnt by the algorithm, as implemented in OpenAI Baseline (Dhariwal et al., 2017). For TRPO, due to its proximity to PPO, we copy PPO's hyperparameters if they exist in both algorithms. We then tune the value update step size in $[3e-4, 5e-4, 1e-3]$. For A2C, we keep the original hyperparameters and tune the number of actors in $[32, 128]$ and the number of timesteps for each actor between consecutive policy updates in $[5, 16, 32]$. For SAC, we tune the reward scale from $[5, 20, 100]$.

The detailed hyperparameters used in our baselines for both MuJoCo and RoboSchool are listed in Tables 7-10.

Table 7: Baseline hyperparameter setting for A2C MuJoCo and RoboSchool tasks.

| Hyperparameter | Value |
|---|---|
| Hidden layer size | $64 \times 2$ |
| Sharing policy and value weights | False |
| Number of hidden layers | 2 |
| Rollout timesteps per actor | 5 |
| Number of actors | 1 |
| Step size | $7e-4$, linear decay |
| Max gradient norm | 0.5 |
| Discount factor ($\gamma$) | 0.99 |

| Hyperparameter | Value |
|---|---|
| Hidden layer size | $64 \times 2$ |
| Number of hidden layers | 2 |
| Sharing policy and value weights | False |
| Rollout timesteps per actor | 32 |
| Number of actors | 32 (Humanoid, Atlas) 128 (Flagrun) |
| Step size | $7e-4$, linear decay |
| Max gradient norm | 0.5 |
| Discount factor ($\gamma$) | 0.99 |

Table 8: Baseline hyperparameter setting for TRPO Mujoco and RoboSchool tasks. The original OpenAI implementation does not support multiple actors sampling trajectories at the same time, so we modified the code to support this feature and accelerate training.

| Hyperparameter | Value | Hyperparameter | Value |
|---|---|---|---|
| Hidden layer size | $32 \times 2$ | Hidden layer size | $64 \times 2$ |
| Number of hidden layers | 2 | Number of hidden layers | 2 |
| Sharing policy and value weights | False | Sharing policy and value weights | False |
| Rollout timesteps per actor | 1024 | Rollout timesteps per actor | 512 |
| Number of actors | 1 | Number of actors | 32 (Humanoid, Atlas) 128 (Flagrun) |
| Value network step size | $1e-3$, constant | value network step size | $1e-3$, constant |
| Max KL divergence | 0.01 | Max KL divergence | 0.01 |
| Discount factor ($\gamma$) | 0.99 | Discount factor ($\gamma$) | 0.99 |
| GAE parameter ($\lambda$) | 0.98 | GAE parameter ($\lambda$) | 0.98 |
| Conjugate gradient damping | 0.1 | Conjugate gradient damping | 0.1 |
| Conjugate gradient iterations | 10 | Conjugate gradient iterations | 10 |
| Value network optimization epochs | 10 | Value network optimization epochs | 15 |
| Value network update minibatch size | 64 | Value network update minibatch size | 4096 |
| Probability ratio clipping range | 0.2 | Probability ratio clipping range | 0.2 |

Table 9: Baseline hyperparameter setting for PPO MuJoCo and RoboSchool tasks.

| Hyperparameter | Value | Hyperparameter | Value |
|---|---|---|---|
| Hidden layer size | $64 \times 2$ | Hidden layer size | $64 \times 2$ |
| Number of hidden layers | 2 | Number of hidden layers | 2 |
| Sharing policy and value weights | False | Sharing policy and value weights | False |
| Rollout timesteps per actor | 2048 | Rollout timesteps per actor | 512 |
| Number of actors | 1 | Number of actors | 32 (Humanoid, Atlas) 128 (Flagrun) |
| Number of minibatches | 32 | Minibatch size | 4096 |
| Step size | $3e-4$, linear decay | Step size | $3e-4$, linear decay |
| Max gradient norm | 0.5 | Max gradient norm | 0.5 |
| Discount factor ($\gamma$) | 0.99 | Discount factor ($\gamma$) | 0.99 |
| GAE parameter ($\lambda$) | 0.95 | GAE parameter ($\lambda$) | 0.95 |
| Number of optimization epochs | 10 | Number of optimization epochs | 15 |
| Probability ratio clipping range | 0.2 | Probability ratio clipping range | 0.2 |

Table 10: Baseline hyperparameter setting for SAC.

| Hyperparameter | Value |
|---|---|
| Hidden layer size | $256 \times 2$ |
| Number of hidden layers | 2 |
| Samples per batch | 256 |
| Replay buffer size | $10^6$ |
| Learning rate | $3e-4$ constant |
| Discount factor ($\gamma$) | 0.99 |
| Target smoothing coefficient ($\tau$) | 0.005 |
| Target update interval | 1 |
| Reward Scaling | 5 (Hopper, Walker, HalfCheetah, Ant) 20 (MuJoCo Humanoid and all RoboSchool tasks) 100 (HumanoidStandup) |

## E  HYPERPARAMETER SAMPLING DETAILS

In Section 4, we present results based on five hyperparameter settings. To obtain such hyperparameter variations, we consider varying the learning rates and the hyperparameters that each algorithm is very sensitive to. For A2C, TRPO, and PPO, we consider a range of rollout timesteps between consecutive policy updates by varying the number of actors or the number of trajectory sampling timesteps for each actor. For SAC, we consider a range of reward scale and a range of target smoothing coefficient.

More concretely, for A2C, we sample the learning rate from $[2e-4, 7e-4, 2e-3]$ linear decay, the number of trajectory sampling timesteps (nsteps) for each actor from $[3, 5, 16, 32]$, and the number of actors (nenvs) from $[1, 4]$. For TRPO, we sample the learning rate of value network (vf_stepsize) from $[3e-4, 5e-4, 1e-3]$ and the number of trajectory sampling timesteps for each actor (nsteps) in $[1024, 2048, 4096, 8192]$. The policy update uses conjugate gradient descent and is controlled by the max KL divergence. For PPO, we sample the learning rate from $[1e-4 \text{ linear}, 3e-4 \text{ constant}]$, the number of actors (nenvs) from $[1, 2, 4, 8]$, and the probability ratio clipping range (cliprange) in $[0.1, 0, 2]$. For SAC, we sample the learning rate from $[1e-4, 3e-4, 1e-3]$ the target smoothing coefficient ($\tau$) from $[0.001, 0.005, 0.01]$, and the reward scale from small, default, and large mode. The default reward scale of 5 is changed to $(3, 5, 20)$; 20 to $(4, 20, 100)$; 100 to $(20, 100, 400)$ for each mode, respectively. Sampled hyperparameters 1-5 for each algorithms are listed in Table 11-14.

Table 11: Sampled hyperparameter settings for A2C.

|               | Learning rate | Nsteps | Nenvs |
|---------------|---------------|--------|-------|
| Baseline      | $7e-4$        | 5      | 1     |
| Hyperparam. 1 | $2e-3$        | 32     | 4     |
| Hyperparam. 2 | $2e-3$        | 32     | 1     |
| Hyperparam. 3 | $7e-4$        | 16     | 1     |
| Hyperparam. 4 | $7e-4$        | 32     | 4     |
| Hyperparam. 5 | $2e-4$        | 3      | 4     |

Table 12: Sampled hyperparameter settings for TRPO.

|               | Vf_stepsize | Nsteps |
|---------------|-------------|--------|
| Baseline      | $1e-3$      | 1024   |
| Hyperparam. 1 | $5e-4$      | 8192   |
| Hyperparam. 2 | $1e-3$      | 4096   |
| Hyperparam. 3 | $3e-4$      | 2048   |
| Hyperparam. 4 | $5e-4$      | 1024   |
| Hyperparam. 5 | $5e-4$      | 4096   |

Table 13: Sampled hyperparameter settings for PPO

|               | Learning rate      | Nenvs | Cliprange |
|---------------|--------------------|-------|-----------|
| Baseline      | $3e-4$ linear      | 1     | 0.2       |
| Hyperparam. 1 | $3e-4$ linear      | 8     | 0.2       |
| Hyperparam. 2 | $1e-4$ constant    | 8     | 0.2       |
| Hyperparam. 3 | $3e-4$ linear      | 4     | 0.1       |
| Hyperparam. 4 | $1e-4$ constant    | 2     | 0.2       |
| Hyperparam. 5 | $3e-4$ linear      | 1     | 0.1       |

Table 14: Sampled hyperparameter settings for SAC

|  | Learning rate | $\tau$ | Mode |
|---|---|---|---|
| Baseline | $3e-4$ | 0.005 | default |
| Hyperparam. 1 | $3e-4$ | 0.005 | small |
| Hyperparam. 2 | $1e-4$ | 0.001 | large |
| Hyperparam. 3 | $1e-3$ | 0.005 | small |
| Hyperparam. 4 | $3e-4$ | 0.01 | small |
| Hyperparam. 5 | $1e-3$ | 0.005 | default |

## F  STATISTICAL SIGNIFICANCE TEST OF $z$-SCORES

For each regularization method, we collect the $z$-scores produced by all seeds and all environments of a certain difficulty (e.g. for $L_2$ on PPO and hard environments, we have 6 envs $\times$ 5 seeds = 30 $z$-scores), and perform Welch's $t$-test (two-sample $t$-test with unequal variance) with the corresponding $z$-scores produced by the baseline. The resulting $p$-values for Table 2 in Section 3 and Table 3 in Section 4 are presented in Table 15 and Table 16, respectively. Note that whether the significance indicates improvement or harm depends on the relative mean $z$-score in Table 2 and Table 3. For example, for BN and dropout in on-policy algorithms, the statistical significance denotes harm, and in most other cases it denotes improvement. From the results, we observe that the improvement is statistically significant ($p < 0.05$) for hard tasks in general, with only a few exceptions. In total, $L_2$, $L_1$, entropy and weight clipping are all statistically significantly better than baseline. For Welch's $t$-test between entropy regularization and other regularizers, see Appendix I.

Table 15: $P$-values from Welch's $t$-test comparing the $z$-scores of regularization methods and baseline.

| Reg \Alg | A2C | | | TRPO | | | PPO | | | SAC | | | TOTAL | | |
|---|---|---|---|---|---|---|---|---|---|---|---|---|---|---|---|
| | Easy | Hard | Total | Easy | Hard | Total | Easy | Hard | Total | Easy | Hard | Total | Easy | Hard | Total |
| Entropy | 0.01 | 0.00 | 0.00 | 0.65 | 0.16 | 0.47 | 0.61 | 0.21 | 0.22 | 0.26 | 0.24 | 0.10 | 0.05 | 0.00 | 0.00 |
| $L_2$ | 0.37 | 0.00 | 0.00 | 0.43 | 0.05 | 0.05 | 0.86 | 0.00 | 0.00 | 0.21 | 0.01 | 0.01 | 0.10 | 0.00 | 0.00 |
| $L_1$ | 0.54 | 0.00 | 0.02 | 0.92 | 0.00 | 0.01 | 0.94 | 0.00 | 0.00 | 0.35 | 0.25 | 0.14 | 0.62 | 0.00 | 0.00 |
| Weight Clip | 0.78 | 0.02 | 0.09 | 0.99 | 0.02 | 0.07 | 0.79 | 0.00 | 0.00 | 0.76 | 0.21 | 0.41 | 0.87 | 0.00 | 0.00 |
| Dropout | 0.00 | 0.00 | 0.00 | N/A | N/A | N/A | 0.27 | 0.80 | 0.74 | 0.21 | 0.00 | 0.00 | 0.02 | 0.43 | 0.05 |
| BatchNorm | 0.00 | 0.00 | 0.00 | 0.00 | 0.00 | 0.00 | 0.00 | 0.10 | 0.00 | 0.38 | 0.02 | 0.25 | 0.00 | 0.00 | 0.00 |

Table 16: $P$-values from Welch's $t$-test comparing the $z$-scores of regularization and baseline, under five sampled hyperparameter settings.

| Reg \Alg | A2C | | | TRPO | | | PPO | | | SAC | | | TOTAL | | |
|---|---|---|---|---|---|---|---|---|---|---|---|---|---|---|---|
| | Easy | Hard | Total | Easy | Hard | Total | Easy | Hard | Total | Easy | Hard | Total | Easy | Hard | Total |
| Entropy | 0.58 | 0.00 | 0.00 | 0.71 | 0.14 | 0.19 | 0.23 | 0.36 | 0.98 | 0.28 | 0.40 | 0.18 | 0.96 | 0.00 | 0.01 |
| $L_2$ | 0.00 | 0.00 | 0.00 | 0.12 | 0.00 | 0.00 | 0.31 | 0.00 | 0.00 | 0.89 | 0.00 | 0.01 | 0.99 | 0.00 | 0.00 |
| $L_1$ | 0.77 | 0.00 | 0.00 | 0.40 | 0.00 | 0.00 | 0.55 | 0.00 | 0.00 | 0.51 | 0.04 | 0.05 | 0.25 | 0.00 | 0.00 |
| Weight Clip | 0.77 | 0.00 | 0.00 | 0.01 | 0.01 | 0.00 | 0.51 | 0.00 | 0.00 | 0.02 | 0.14 | 0.69 | 0.37 | 0.00 | 0.00 |
| Dropout | 0.00 | 0.00 | 0.00 | N/A | N/A | N/A | 0.00 | 0.00 | 0.00 | 0.96 | 0.29 | 0.41 | 0.00 | 0.00 | 0.00 |
| BatchNorm | 0.00 | 0.00 | 0.00 | 0.00 | 0.00 | 0.00 | 0.00 | 0.00 | 0.00 | 0.39 | 0.05 | 0.03 | 0.00 | 0.00 | 0.00 |

## G  $z$-SCORE STATISTICS UNDER MORE RANDOM SEEDS ON MUJOCO

To further verify our result, we increase the number of seeds from 5 to 10 and present $z$-scores for the six MuJoCo environments (easy: Hopper, Walker, HalfCheetah; hard: Ant, Humanoid, HumanoidStandup) in Table 17. We also present tests of statistical significance in Table 18. Our observations are consistent with those in Table 2. Due to the large computation cost required, we do not include the three hard Roboschool environments in the calculation of $z$-scores.

Table 17: Average $z$-scores comparing regularizers vs. baseline on MuJoCo under the default hyperparameter setting, where experiments in each environment are conducted over 10 random seeds.

| Reg \Alg | A2C | | | TRPO | | | PPO | | | SAC | | | TOTAL | | |
|---|---|---|---|---|---|---|---|---|---|---|---|---|---|---|---|
| | Easy | Hard | Total | Easy | Hard | Total | Easy | Hard | Total | Easy | Hard | Total | Easy | Hard | Total |
| Baseline | 0.34 | -0.73 | -0.19 | 0.26 | 0.11 | 0.18 | 0.32 | -1.36 | -0.52 | -0.21 | -0.35 | -0.28 | 0.18 | -0.58 | -0.20 |
| Entropy | **1.15** | 0.94 | **1.05** | 0.20 | 0.35 | 0.28 | **0.40** | -1.06 | -0.33 | 0.29 | 0.23 | 0.26 | **0.51** | 0.12 | 0.31 |
| $L_2$ | 0.50 | **1.27** | 0.88 | **0.54** | 0.46 | **0.50** | 0.25 | 0.69 | 0.47 | **0.33** | **0.38** | **0.35** | 0.40 | **0.70** | **0.55** |
| $L_1$ | 0.19 | 0.56 | 0.38 | 0.34 | 0.46 | 0.40 | 0.26 | **0.79** | **0.52** | 0.13 | -0.42 | -0.15 | 0.23 | 0.34 | 0.29 |
| Weight Clip | 0.16 | 0.09 | 0.12 | 0.19 | **0.71** | 0.45 | 0.31 | 0.77 | 0.54 | -0.38 | 0.12 | -0.13 | 0.07 | 0.42 | 0.25 |
| Dropout | -1.16 | -0.98 | -1.07 | N/A | N/A | N/A | -0.10 | 0.57 | 0.24 | **0.33** | 0.22 | 0.28 | -0.31 | -0.06 | -0.19 |
| BatchNorm | -1.19 | -1.15 | -1.17 | -1.53 | -2.08 | -1.81 | -1.43 | -0.40 | -0.92 | -0.49 | -0.18 | -0.33 | -1.16 | -0.95 | -1.06 |

Table 18: $p$-values from Welch's $t$-test comparing the $z$-scores of regularization methods and baseline under the default hyperparameter setting and 10 random seeds.

| Reg \Alg | A2C | | | TRPO | | | PPO | | | SAC | | | TOTAL | | |
|---|---|---|---|---|---|---|---|---|---|---|---|---|---|---|---|
| | Easy | Hard | Total | Easy | Hard | Total | Easy | Hard | Total | Easy | Hard | Total | Easy | Hard | Total |
| Entropy | 0.00 | 0.00 | 0.00 | 0.85 | 0.00 | 0.27 | 0.67 | 0.00 | 0.30 | 0.21 | 0.03 | 0.02 | 0.02 | 0.00 | 0.00 |
| $L_2$ | 0.52 | 0.00 | 0.00 | 0.13 | 0.00 | 0.00 | 0.86 | 0.00 | 0.00 | 0.10 | 0.02 | 0.00 | 0.06 | 0.00 | 0.00 |
| $L_1$ | 0.29 | 0.00 | 0.00 | 0.73 | 0.00 | 0.03 | 0.87 | 0.00 | 0.00 | 0.25 | 0.57 | 0.23 | 0.66 | 0.00 | 0.00 |
| Weight Clip | 0.44 | 0.00 | 0.01 | 0.94 | 0.00 | 0.01 | 0.80 | 0.00 | 0.00 | 0.74 | 0.10 | 0.32 | 0.54 | 0.00 | 0.00 |
| Dropout | 0.00 | 0.00 | 0.00 | N/A | N/A | N/A | 0.08 | 0.00 | 0.00 | 0.08 | 0.05 | 0.01 | 0.00 | 0.00 | 0.70 |
| BatchNorm | 0.00 | 0.00 | 0.00 | 0.00 | 0.00 | 0.00 | 0.00 | 0.00 | 0.03 | 0.32 | 0.37 | 0.94 | 0.00 | 0.01 | 0.00 |

# H    IMPROVEMENT PERCENTAGE FOR HYPERPARAMETER EXPERIMENTS

We provide the percentage of improvement result in Table 19 as a complement with Table 3, for the experiments with multiple sampled hyperparameters.

Table 19: Percentage (%) of environments where the final performance "improves" when using regularization, under five randomly sampled training hyperparameters for each algorithm.

| Reg \ Alg | A2C | | | TRPO | | | PPO | | | SAC | | | TOTAL | | |
|---|---|---|---|---|---|---|---|---|---|---|---|---|---|---|---|
| | Easy | Hard | Total | Easy | Hard | Total | Easy | Hard | Total | Easy | Hard | Total | Easy | Hard | Total |
| Entropy | **20.0** | 40.0 | 32.0 | 0.0 | 26.7 | 16.0 | 10.0 | 33.3 | 24.0 | **60.0** | 13.3 | **32.0** | **22.5** | 28.3 | 26.0 |
| $L_2$ | **20.0** | 60.0 | **44.0** | 10.0 | 40.0 | 28.0 | **20.0** | 86.7 | **60.0** | 10.0 | **40.0** | 28.0 | 15.0 | **56.7** | **40.0** |
| $L_1$ | 10.0 | 53.3 | 36.0 | 10.0 | **46.7** | 32.0 | 10.0 | **86.7** | 56.0 | 20.0 | 26.7 | 24.0 | 12.5 | 53.3 | 37.0 |
| Weight Clip | 0.0 | 46.7 | 28.0 | **40.0** | **46.7** | **44.0** | 10.0 | 73.3 | 48.0 | 0.0 | 33.3 | 20.0 | 12.5 | 50.0 | 35.0 |
| Dropout | **20.0** | 0.0 | 8.0 | N/A | N/A | N/A | 0.0 | 40.0 | 24.0 | 0.0 | 20.0 | 12.0 | 6.7 | 20.0 | 14.7 |
| BatchNorm | 0.0 | 0.0 | 0.0 | 10.0 | 0.0 | 4.0 | 10.0 | 33.3 | 24.0 | 20.0 | 20.0 | 20.0 | 10.0 | 13.3 | 12.0 |

# I STATISTICAL SIGNIFICANCE TEST OF $z$-SCORES (ENTROPY REGULARIZATION)

As a complement to Table 2 in Section 3 and Table 3 in Section 4, we present the $p$-value results from Welch's $t$-test comparing the $z$-scores of entropy regularization with other regularizers in Table 20 and Table 21. Note that whether the significance indicates improvement or harm over entropy regularization depends on the relative mean $z$-score in Table 2 under default hyperparameter setting and Table 3 under sampled hyperparameter setting. We observe that in total, $L_2$ has significant improvement over entropy in both default hyperparameter setting and sampled hyperparameter setting. $L_1$ and weight clipping are significantly better than entropy under sampled hyperparameter setting. In general, the improvement over entropy is statistically more significant for hard tasks.

Table 20: $P$-values from Welch's $t$-test comparing the $z$-scores of entropy regularization and other regularizers, under the default hyperparameter setting.

| Reg \Alg | A2C | | | TRPO | | | PPO | | | SAC | | | TOTAL | | |
|---|---|---|---|---|---|---|---|---|---|---|---|---|---|---|---|
| | Easy | Hard | Total | Easy | Hard | Total | Easy | Hard | Total | Easy | Hard | Total | Easy | Hard | Total |
| $L_2$ | 0.05 | 0.56 | 0.07 | 0.22 | 0.59 | 0.21 | 0.66 | 0.00 | 0.00 | 0.89 | 0.07 | 0.12 | 0.58 | 0.00 | 0.02 |
| $L_1$ | 0.00 | 0.00 | 0.00 | 0.57 | 0.05 | 0.07 | 0.62 | 0.00 | 0.00 | 0.67 | 0.93 | 0.75 | 0.06 | 0.10 | 0.83 |
| Weight Clip | 0.01 | 0.00 | 0.00 | 0.64 | 0.27 | 0.25 | 0.79 | 0.00 | 0.00 | 0.05 | 0.81 | 0.37 | 0.02 | 0.42 | 0.42 |
| Dropout | 0.00 | 0.00 | 0.00 | N/A | N/A | N/A | 0.07 | 0.43 | 0.12 | 0.93 | 0.01 | 0.03 | 0.00 | 0.00 | 0.00 |
| BatchNorm | 0.00 | 0.00 | 0.00 | 0.00 | 0.00 | 0.00 | 0.00 | 0.00 | 0.00 | 0.01 | 0.16 | 0.60 | 0.00 | 0.00 | 0.00 |

Table 21: $P$-values from Welch's $t$-test comparing the $z$-scores of entropy regularization and other regularizers, under five sampled hyperparameter settings for each policy optimization algorithm.

| Reg \Alg | A2C | | | TRPO | | | PPO | | | SAC | | | TOTAL | | |
|---|---|---|---|---|---|---|---|---|---|---|---|---|---|---|---|
| | Easy | Hard | Total | Easy | Hard | Total | Easy | Hard | Total | Easy | Hard | Total | Easy | Hard | Total |
| $L_2$ | 0.01 | 0.00 | 0.59 | 0.25 | 0.00 | 0.01 | 0.02 | 0.00 | 0.00 | 0.33 | 0.01 | 0.17 | 0.96 | 0.00 | 0.00 |
| $L_1$ | 0.41 | 0.04 | 0.04 | 0.69 | 0.00 | 0.01 | 0.07 | 0.00 | 0.00 | 0.63 | 0.20 | 0.53 | 0.23 | 0.00 | 0.00 |
| Weight Clip | 0.76 | 0.86 | 0.96 | 0.03 | 0.11 | 0.01 | 0.59 | 0.00 | 0.00 | 0.00 | 0.45 | 0.11 | 0.40 | 0.00 | 0.04 |
| Dropout | 0.00 | 0.00 | 0.00 | N/A | N/A | N/A | 0.00 | 0.00 | 0.00 | 0.31 | 0.85 | 0.56 | 0.00 | 0.00 | 0.00 |
| BatchNorm | 0.00 | 0.00 | 0.00 | 0.00 | 0.00 | 0.00 | 0.00 | 0.00 | 0.00 | 0.80 | 0.17 | 0.33 | 0.00 | 0.00 | 0.00 |

## J  ADDITIONAL METRICS

### J.1  RANKING ALL REGULARIZERS

We compute the "average ranking" metric to compare the relative effectiveness of different regularization methods. Note that the average ranking of different methods across a set of tasks/datasets has been adopted as a metric before, as in (Ranftl et al., 2019) and (Knapitsch et al., 2017). Here, we rank the performance of all the regularization methods, together with the baseline, for each algorithm and task, and present the average ranks in Table 22 and Table 23, with statistical significance tests in Table 24 and 25. The ranks of returns among different regularizers are collected for each environment (after averaging over 5 random seeds), and then the mean rank over all seeds is calculated. From Table 22 and Table 23, we observe that, except for BN and dropout in on-policy algorithms, all regularizations on average outperform baselines. Again, $L_2$ regularization is the strongest in most cases. Other similar observations can be made as in previous tables. For every algorithm, baseline ranks lower on harder tasks than easier ones; in total, it ranks 3.50 for easier tasks and 5.25 for harder tasks. This indicates that regularization is more effective when the tasks are harder.

Table 22: The average rank in the mean return for different regularization methods under default hyperparameter settings. $L_2$ regularization tops the ranking for most algorithms and environment difficulties.

| Reg \ Alg | A2C | | | TRPO | | | PPO | | | SAC | | | TOTAL | | |
|---|---|---|---|---|---|---|---|---|---|---|---|---|---|---|---|
| | Easy | Hard | Total | Easy | Hard | Total | Easy | Hard | Total | Easy | Hard | Total | Easy | Hard | Total |
| Baseline | 3.33 | 4.50 | 4.11 | 3.33 | 4.67 | 4.22 | 3.00 | 6.00 | 5.00 | 4.33 | 5.83 | 5.33 | 3.50 | 5.25 | 4.67 |
| Entropy | **1.00** | **1.50** | **1.33** | 4.67 | 3.00 | 3.56 | **3.00** | 4.17 | 3.78 | **3.00** | 3.83 | 3.55 | 2.92 | 3.13 | 3.06 |
| $L_2$ | 2.67 | **1.50** | 1.89 | **1.33** | 2.83 | **2.33** | **3.00** | **2.17** | **2.45** | **3.00** | 2.67 | **2.78** | **2.50** | **2.29** | **2.36** |
| $L_1$ | 4.33 | 3.67 | 3.89 | 2.67 | **2.17** | 2.34 | 3.33 | 2.67 | 2.89 | 3.67 | 4.83 | 4.44 | 3.50 | 3.34 | 3.39 |
| Weight Clip | 3.67 | 3.83 | 3.78 | 3.00 | 2.33 | 2.55 | **3.00** | 2.50 | 2.67 | 4.33 | 4.17 | 4.22 | 3.50 | 3.21 | 3.31 |
| Dropout | 6.00 | 6.00 | 6.00 | N/A | N/A | N/A | 5.67 | 4.67 | 5.00 | 3.33 | 3.17 | 3.22 | 5.00 | 4.61 | 4.74 |
| BatchNorm | 7.00 | 7.00 | 7.00 | 6.00 | 6.00 | 6.00 | 7.00 | 5.83 | 6.22 | 6.33 | 3.50 | 4.44 | 6.58 | 5.58 | 5.92 |

Table 23: The average rank in the mean return for different regularization methods, under five randomly sampled training hyperparameters for each algorithm.

| Reg \ Alg | A2C | | | TRPO | | | PPO | | | SAC | | | TOTAL | | |
|---|---|---|---|---|---|---|---|---|---|---|---|---|---|---|---|
| | Easy | Hard | Total | Easy | Hard | Total | Easy | Hard | Total | Easy | Hard | Total | Easy | Hard | Total |
| Baseline | **2.70** | 4.13 | 3.65 | 3.70 | 3.40 | 3.50 | 3.00 | 5.53 | 4.69 | 4.20 | 5.00 | 4.73 | 3.40 | 4.52 | 4.14 |
| Entropy | 3.50 | 2.93 | 3.12 | 3.60 | 3.47 | 3.51 | 4.30 | 4.40 | 4.37 | **3.10** | 4.47 | 4.01 | 3.63 | 3.82 | 3.75 |
| $L_2$ | 4.40 | **2.27** | 2.98 | 2.50 | 2.53 | **2.52** | **1.90** | **1.80** | **1.83** | 3.50 | **2.73** | **2.99** | **3.08** | **2.33** | **2.58** |
| $L_1$ | **2.70** | 2.53 | **2.59** | 3.10 | **2.27** | 2.55 | 2.80 | 2.20 | 2.40 | 3.70 | 4.00 | 3.90 | **3.08** | 2.75 | 2.86 |
| Weight Clip | 3.30 | 3.13 | 3.19 | **2.20** | 3.33 | 2.95 | 3.70 | 2.87 | 3.15 | 5.80 | 4.27 | 4.78 | 3.75 | 3.40 | 3.52 |
| Dropout | 4.40 | 6.07 | 5.51 | N/A | N/A | N/A | 6.10 | 5.33 | 5.59 | 4.20 | 4.27 | 4.25 | 4.90 | 5.22 | 5.12 |
| BatchNorm | 7.00 | 6.93 | 6.95 | 5.90 | 6.00 | 5.97 | 6.20 | 5.80 | 5.93 | 3.50 | 3.27 | 3.35 | 5.65 | 5.50 | 5.55 |

Table 24: *P*-values from Welch's *t*-test comparing the average rank of regularization and baseline, under the default hyperparmeter setting.

| Reg \Alg | A2C | | | TRPO | | | PPO | | | SAC | | | TOTAL | | |
|---|---|---|---|---|---|---|---|---|---|---|---|---|---|---|---|
| | Easy | Hard | Total | Easy | Hard | Total | Easy | Hard | Total | Easy | Hard | Total | Easy | Hard | Total |
| Entropy | 0.12 | 0.00 | 0.00 | 0.38 | 0.07 | 0.41 | 1.00 | 0.02 | 0.06 | 0.63 | 0.25 | 0.18 | 0.46 | 0.00 | 0.00 |
| $L_2$ | 0.69 | 0.00 | 0.01 | 0.07 | 0.01 | 0.00 | 1.00 | 0.00 | 0.03 | 0.60 | 0.07 | 0.06 | 0.21 | 0.00 | 0.00 |
| $L_1$ | 0.58 | 0.22 | 0.75 | 0.53 | 0.00 | 0.00 | 0.91 | 0.00 | 0.08 | 0.67 | 0.28 | 0.21 | 1.00 | 0.00 | 0.00 |
| Weight Clip | 0.67 | 0.29 | 0.47 | 0.87 | 0.05 | 0.09 | 1.00 | 0.01 | 0.05 | 1.00 | 0.28 | 0.42 | 1.00 | 0.00 | 0.01 |
| Dropout | 0.09 | 0.01 | 0.00 | N/A | N/A | N/A | 0.29 | 0.35 | 1.00 | 0.73 | 0.02 | 0.05 | 0.23 | 0.21 | 0.90 |
| BatchNorm | 0.05 | 0.00 | 0.00 | 0.09 | 0.01 | 0.00 | 0.12 | 0.85 | 0.25 | 0.37 | 0.07 | 0.44 | 0.00 | 0.51 | 0.01 |

Table 25: *P*-values from Welch's *t*-test comparing the average rank of regularization and baseline, under the 5 randomly sampled hyperparmeter settings.

| Reg \Alg | A2C | | | TRPO | | | PPO | | | SAC | | | TOTAL | | |
|---|---|---|---|---|---|---|---|---|---|---|---|---|---|---|---|
| | Easy | Hard | Total | Easy | Hard | Total | Easy | Hard | Total | Easy | Hard | Total | Easy | Hard | Total |
| Entropy | 0.31 | 0.04 | 0.31 | 0.91 | 0.45 | 0.54 | 0.10 | 0.08 | 0.74 | 0.39 | 0.43 | 0.23 | 0.73 | 0.01 | 0.10 |
| $L_2$ | 0.09 | 0.00 | 0.43 | 0.01 | 0.02 | 0.00 | 0.33 | 0.00 | 0.00 | 0.28 | 0.04 | 0.02 | 0.34 | 0.00 | 0.00 |
| $L_1$ | 0.89 | 0.04 | 0.10 | 0.49 | 0.01 | 0.01 | 0.45 | 0.00 | 0.00 | 0.63 | 0.19 | 0.18 | 0.35 | 0.00 | 0.00 |
| Weight Clip | 0.65 | 0.09 | 0.30 | 0.06 | 0.60 | 0.10 | 0.21 | 0.00 | 0.03 | 0.04 | 0.39 | 0.74 | 0.51 | 0.00 | 0.03 |
| Dropout | 0.08 | 0.00 | 0.00 | 0.00 | 0.00 | 0.00 | 0.00 | 1.00 | 0.02 | 1.00 | 0.27 | 0.39 | 0.00 | 0.25 | 0.00 |
| BatchNorm | 0.00 | 0.00 | 0.00 | 0.01 | 0.00 | 0.00 | 0.00 | 0.40 | 0.00 | 0.39 | 0.04 | 0.03 | 0.00 | 0.01 | 0.00 |

## J.2 SCALED RETURNS

Min-max scaling is a linear-mapping operation to map values ranging from $[\min(x), \max(x)]$ to $[0, 1]$, using $x' = \frac{x - \min(x)}{\max(x) - \min(x)}$. For each environment and policy optimization algorithm (for example, PPO on Ant), we calculate a "scaled return" for each regularization method and the baseline, using the maximum mean return obtained using any regularization method (including baseline) as $\max(x)$ and 0 as $\min(x)$, on positively clipped returns. We then average the scaled returns of mean return over environments of a certain difficulty (easy/hard). We present the results under the default hyperparameter setting in Table 26-28 and the results under sampled hyperparameter settings in Table 29-31. To analyze whether regularization significantly improves over the baseline and whether conventional regularizers significantly improves over entropy, we perform Welch's *t*-test on the scaled returns, using an identical approach to the one we used for $z$-score. Our observation is similar to the ones we made in Section 3 and Section 4.

Table 26: Scaled returns for each regularization method under the default hyperparameter setting.

| Reg \Alg | A2C | | | TRPO | | | PPO | | | SAC | | | TOTAL | | |
|---|---|---|---|---|---|---|---|---|---|---|---|---|---|---|---|
| | Easy | Hard | Total | Easy | Hard | Total | Easy | Hard | Total | Easy | Hard | Total | Easy | Hard | Total |
| Entropy | **100.0** | **93.0** | **95.3** | 86.2 | 84.3 | 85.0 | **95.2** | 57.2 | 69.9 | 94.9 | 89.0 | 91.0 | **94.1** | 80.9 | 85.3 |
| $L_2$ | 74.8 | 90.2 | 85.1 | **97.2** | 87.2 | 90.6 | 88.0 | 93.2 | 91.5 | **97.1** | 92.4 | 93.9 | 89.3 | **90.7** | **90.3** |
| $L_1$ | 58.8 | 70.6 | 66.7 | 91.9 | **95.5** | **94.3** | 90.0 | **93.5** | **92.3** | 95.0 | 89.7 | 91.5 | 83.6 | 87.3 | 86.2 |
| Weight Clip | 61.2 | 65.3 | 63.9 | 90.7 | 89.6 | 90.0 | 92.7 | 88.4 | 89.8 | 91.6 | 89.2 | 90.0 | 84.0 | 83.1 | 83.4 |
| Dropout | 0.85 | 9.05 | 6.32 | N/A | N/A | N/A | 76.2 | 42.5 | 53.7 | 97.0 | **96.2** | **96.5** | 58.0 | 49.3 | 52.2 |
| BatchNorm | 0.00 | 6.32 | 4.21 | 21.8 | 12.9 | 15.9 | 25.7 | 30.7 | 29.1 | 88.2 | 92.9 | 91.4 | 33.9 | 35.7 | 35.1 |
| Baseline | 64.1 | 48.9 | 54.0 | 91.8 | 77.9 | 82.5 | 89.3 | 47.9 | 61.7 | 90.5 | 86.4 | 87.8 | 83.9 | 65.3 | 71.5 |

Table 27: *P*-values from Welch's *t*-test comparing the scaled returns of regularization and baseline, under the default hyperparmeter setting.

| Reg \Alg | A2C | | | TRPO | | | PPO | | | SAC | | | TOTAL | | |
|---|---|---|---|---|---|---|---|---|---|---|---|---|---|---|---|
| | Easy | Hard | Total | Easy | Hard | Total | Easy | Hard | Total | Easy | Hard | Total | Easy | Hard | Total |
| Entropy | 0.01 | 0.00 | 0.00 | 0.64 | 0.33 | 0.66 | 0.67 | 0.34 | 0.36 | 0.23 | 0.54 | 0.25 | 0.07 | 0.00 | 0.00 |
| $L_2$ | 0.38 | 0.00 | 0.00 | 0.50 | 0.16 | 0.14 | 0.92 | 0.00 | 0.00 | 0.11 | 0.07 | 0.02 | 0.27 | 0.00 | 0.00 |
| $L_1$ | 0.62 | 0.01 | 0.06 | 0.99 | 0.00 | 0.03 | 0.95 | 0.00 | 0.00 | 0.26 | 0.26 | 0.12 | 0.99 | 0.00 | 0.00 |
| Weight Clip | 0.80 | 0.04 | 0.14 | 0.94 | 0.10 | 0.22 | 0.80 | 0.00 | 0.00 | 0.80 | 0.46 | 0.44 | 0.98 | 0.00 | 0.00 |
| Dropout | 0.00 | 0.00 | 0.00 | N/A | N/A | N/A | 0.29 | 0.56 | 0.34 | 0.12 | 0.00 | 0.00 | 0.00 | 0.00 | 0.00 |
| BatchNorm | 0.00 | 0.00 | 0.00 | 0.00 | 0.00 | 0.00 | 0.00 | 0.05 | 0.00 | 0.60 | 0.03 | 0.15 | 0.00 | 0.00 | 0.00 |

Table 28: *P*-values from Welch's *t*-test comparing the scaled returns of entropy and other regularizers, under the default hyperparmeter setting.

| Reg \Alg | A2C | | | TRPO | | | PPO | | | SAC | | | TOTAL | | |
|---|---|---|---|---|---|---|---|---|---|---|---|---|---|---|---|
| | Easy | Hard | Total | Easy | Hard | Total | Easy | Hard | Total | Easy | Hard | Total | Easy | Hard | Total |
| $L_2$ | 0.07 | 0.58 | 0.07 | 0.25 | 0.70 | 0.29 | 0.49 | 0.00 | 0.00 | 0.70 | 0.30 | 0.28 | 0.42 | 0.00 | 0.07 |
| $L_1$ | 0.00 | 0.00 | 0.00 | 0.63 | 0.07 | 0.09 | 0.66 | 0.00 | 0.00 | 0.74 | 0.73 | 0.86 | 0.05 | 0.05 | 0.77 |
| Weight Clip | 0.01 | 0.00 | 0.00 | 0.69 | 0.49 | 0.42 | 0.84 | 0.00 | 0.01 | 0.16 | 0.88 | 0.73 | 0.07 | 0.54 | 0.51 |
| Dropout | 0.00 | 0.00 | 0.00 | N/A | N/A | N/A | 0.09 | 0.14 | 0.05 | 0.72 | 0.02 | 0.03 | 0.00 | 0.00 | 0.00 |
| BatchNorm | 0.00 | 0.00 | 0.00 | 0.00 | 0.00 | 0.00 | 0.00 | 0.01 | 0.00 | 0.04 | 0.22 | 0.91 | 0.00 | 0.00 | 0.00 |

Table 29: Scaled returns for each regularization method under the five sampled hyperparameter settings.

| Reg \Alg | A2C | | | TRPO | | | PPO | | | SAC | | | TOTAL | | |
|---|---|---|---|---|---|---|---|---|---|---|---|---|---|---|---|
| | Easy | Hard | Total | Easy | Hard | Total | Easy | Hard | Total | Easy | Hard | Total | Easy | Hard | Total |
| Baseline | 86.8 | 60.8 | 71.2 | 87.9 | 88.3 | 88.1 | 93.7 | 66.6 | 77.4 | 92.8 | 86.8 | 89.2 | 90.3 | 75.6 | 81.5 |
| Entropy | 83.8 | 81.2 | 82.2 | 86.8 | 91.0 | 89.3 | 89.8 | 65.8 | 75.4 | **97.5** | 87.5 | 91.5 | 89.5 | 81.4 | 84.6 |
| $L_2$ | 69.9 | **92.8** | 83.6 | 93.6 | 95.4 | **94.7** | **96.4** | **90.6** | **92.9** | 93.3 | **95.5** | **94.6** | 88.3 | **93.6** | **91.5** |
| $L_1$ | **89.1** | 88.7 | **88.8** | 89.9 | **97.1** | 94.2 | 95.2 | 89.0 | 91.5 | 92.7 | 91.7 | 92.1 | **91.7** | 91.6 | **91.7** |
| Weight Clip | 85.5 | 81.3 | 83.0 | 96.4 | 96.6 | 96.5 | 91.3 | 84.1 | 87.0 | 86.7 | 91.6 | 89.6 | 90.0 | 88.4 | 89.0 |
| Dropout | 59.3 | 21.9 | 36.9 | N/A | N/A | N/A | 71.2 | 57.6 | 63.0 | 94.1 | 89.3 | 91.2 | 74.9 | 56.3 | 63.7 |
| BatchNorm | 0.00 | 14.9 | 8.96 | 41.7 | 47.9 | 45.4 | 70.6 | 53.0 | 60.0 | 96.1 | 92.6 | 94.0 | 52.1 | 52.1 | 52.1 |

Table 30: *P*-values from Welch's *t*-test comparing the scaled returns of regularization and baseline, under five sampled hyperparameters.

| Reg \Alg | A2C | | | TRPO | | | PPO | | | SAC | | | TOTAL | | |
|---|---|---|---|---|---|---|---|---|---|---|---|---|---|---|---|
| | Easy | Hard | Total | Easy | Hard | Total | Easy | Hard | Total | Easy | Hard | Total | Easy | Hard | Total |
| Entropy | 0.59 | 0.00 | 0.01 | 0.74 | 0.18 | 0.51 | 0.22 | 0.87 | 0.57 | 0.06 | 0.69 | 0.18 | 0.63 | 0.07 | 0.04 |
| $L_2$ | 0.00 | 0.00 | 0.00 | 0.09 | 0.00 | 0.00 | 0.40 | 0.00 | 0.00 | 0.86 | 0.00 | 0.00 | 0.32 | 0.00 | 0.00 |
| $L_1$ | 0.71 | 0.00 | 0.00 | 0.51 | 0.00 | 0.00 | 0.63 | 0.00 | 0.00 | 0.98 | 0.03 | 0.12 | 0.49 | 0.00 | 0.00 |
| Weight Clip | 0.80 | 0.00 | 0.00 | 0.01 | 0.00 | 0.00 | 0.47 | 0.00 | 0.00 | 0.08 | 0.05 | 0.83 | 0.86 | 0.00 | 0.00 |
| Dropout | 0.00 | 0.00 | 0.00 | N/A | N/A | N/A | 0.00 | 0.03 | 0.00 | 0.66 | 0.26 | 0.26 | 0.00 | 0.00 | 0.00 |
| BatchNorm | 0.00 | 0.00 | 0.00 | 0.00 | 0.00 | 0.00 | 0.00 | 0.00 | 0.00 | 0.19 | 0.02 | 0.01 | 0.00 | 0.00 | 0.00 |

Table 31: *P*-values from Welch's *t*-test comparing the scaled returns of entropy and other regularizers, under five sampled hyperparameters.

| Reg \Alg | A2C | | | TRPO | | | PPO | | | SAC | | | TOTAL | | |
|---|---|---|---|---|---|---|---|---|---|---|---|---|---|---|---|
| | Easy | Hard | Total | Easy | Hard | Total | Easy | Hard | Total | Easy | Hard | Total | Easy | Hard | Total |
| $L_2$ | 0.01 | 0.00 | 0.69 | 0.07 | 0.01 | 0.00 | 0.03 | 0.00 | 0.00 | 0.07 | 0.00 | 0.05 | 0.57 | 0.00 | 0.00 |
| $L_1$ | 0.37 | 0.05 | 0.04 | 0.36 | 0.00 | 0.00 | 0.08 | 0.00 | 0.00 | 0.10 | 0.06 | 0.72 | 0.24 | 0.00 | 0.00 |
| Weight Clip | 0.75 | 0.99 | 0.82 | 0.01 | 0.00 | 0.00 | 0.64 | 0.00 | 0.00 | 0.00 | 0.10 | 0.32 | 0.77 | 0.00 | 0.00 |
| Dropout | 0.00 | 0.00 | 0.00 | N/A | N/A | N/A | 0.00 | 0.07 | 0.00 | 0.17 | 0.47 | 0.87 | 0.00 | 0.00 | 0.00 |
| BatchNorm | 0.00 | 0.00 | 0.00 | 0.00 | 0.00 | 0.00 | 0.00 | 0.01 | 0.00 | 0.52 | 0.05 | 0.14 | 0.00 | 0.00 | 0.00 |

## K  JUSTIFICATION OF METHODOLOGY AND STATISTICAL SIGNIFICANCE

In this section, we provide rigorous justification that, (1) when the sample size is large enough $(n \geq 30)$, the normality assumption for the sampling distribution is not needed (loc); (2) since we test on the entire set of environments instead of on a single environment, our sample size is large enough to satisfy the condition of Welch's $t$-test and provide reliable results.

Consider two distributions with mean and variance pairs $(\mu_1, \sigma_1^2)$ and $(\mu_2, \sigma_2^2)$, respectively, where **neither distribution needs to be normal**, and the mean and variances are unknown. Let $H_0 : \mu_1 = \mu_2$ be the null hypothesis, and $H_1 : \mu_1 \neq \mu_2$ be the alternate hypothesis. Let $(X_1, X_2, \ldots, X_n)$ and $(Y_1, Y_2, \ldots, Y_n)$ be independent samples from the two distributions. Then, under the null hypothesis, the $t$ statistic from Welch's $t$-test converges in distribution to $\mathcal{N}(0,1)$ as $n \to \infty$. We formalize the above statement below.

**Theorem K.1.** *Consider two distributions with mean and variance pairs $(\mu_1, \sigma_1^2)$ and $(\mu_2, \sigma_2^2)$, where the mean and variances are unknown. Define $H_0 : \mu_1 = \mu_2$ and $H_1 : \mu_1 \neq \mu_2$. Let $(X_1, X_2, \ldots, X_n)$ and $(Y_1, Y_2, \ldots, Y_n)$ be independent samples from the two distributions. Then, under $H_0$, the $t$ statistic from Welch's $t$-test converges in distribution to the standard normal distribution as $n \to \infty$. That is, $t_n = \frac{\sqrt{n}(\overline{X}_n - \overline{Y}_n)}{\sqrt{S_{X,n}^2 + S_{Y,n}^2}} \xrightarrow{d} \mathcal{N}(0,1)$, where $\overline{X}_n, \overline{Y}_n$ are the sample means for $(X_1, X_2, \ldots, X_n)$ and $(Y_1, Y_2, \ldots, Y_n)$; $S_{X,n}^2$ and $S_{Y,n}^2$ are the sample variances.*

*Proof.* We have $S_{X,n}^2 \xrightarrow{p} \sigma_1^2$ and $S_{Y,n}^2 \xrightarrow{p} \sigma_2^2$. Then due to independence, $(S_{X,n}^2, S_{Y,n}^2) \xrightarrow{p} (\sigma_1^2, \sigma_2^2)$. By the continuous mapping theorem, $\sqrt{S_{X,n}^2 + S_{Y,n}^2} \xrightarrow{p} \sqrt{\sigma_1^2 + \sigma_2^2}$. The rejection / acceptance region of $t_n$ is based on the null hypothesis. Under the null hypothesis, according to the Central Limit Theorem, $\sqrt{n}(\overline{X}_n - \mu_1) \xrightarrow{d} \mathcal{N}(0, \sigma_1^2)$, $\sqrt{n}(\overline{Y}_n - \mu_1) \xrightarrow{d} \mathcal{N}(0, \sigma_2^2)$. Then due to independence, $(\sqrt{n}(\overline{X}_n - \mu_1), \sqrt{n}(\overline{Y}_n - \mu_1)) \xrightarrow{d} (\mathcal{N}(0, \sigma_1^2), \mathcal{N}(0, \sigma_2^2))$. By Slutsky's theorem, $\sqrt{n}(\overline{X}_n - \overline{Y}_n) \xrightarrow{d} \mathcal{N}(0, \sigma_1^2 + \sigma_2^2)$. Again by Slutsky, $\frac{\sqrt{n}(\overline{X}_n - \overline{Y}_n)}{\sqrt{S_{X,n}^2 + S_{Y,n}^2}} \xrightarrow{d} \mathcal{N}(0,1)$. $\square$

Therefore, if $n \geq 30$ (i.e. the sample size is large), we do not need the normality assumption of our distribution to apply Welch's $t$-test, and we can use our $t$-statistic to obtain the $p$-value the same way as from the $z$-test (loc) (i.e. the $p$-value equals $2 \cdot \Phi(-|t|)$, where $\Phi$ is the cumulative distribution function (CDF) of the standard normal distribution). Also, the $t$-test can be applied when $n$ grows much larger than 30 (lar).

We now show that our sample size is large enough to apply the above theorem. For each algorithm and regularizer, we calculate the average $z$-score, the average ranking, and the average scaled return over a set of environments and all seeds. We then test whether the performance of a regularizer is significantly different from that of baseline. We take the average $z$-score metric as an example. Let $E$ be the set of environments with uniform distribution over the environments, and let $S$ be the set of seeds. For $e \in E$ and $s \in S$, let $f_{\text{reg}}(e, s)$ denote the $z$-score of a certain regularizer under an environment $e$ and seed $s$, and let $f_{\text{baseline}}(e, s)$ denote the $z$-score under the baseline. We use Welch's $t$-test to test whether $\mu_{\text{reg}} \neq \mu_{\text{baseline}}$, given unknown $\sigma_{\text{reg}}, \sigma_{\text{baseline}}$, on a policy optimization algorithm.

For experiments in Section 3 (e.g. Table 2), for each policy optimization algorithm, we test the distribution of $\{f_{\text{reg}}(e, s) : e \in E, s \in S\}$ versus $\{f_{\text{baseline}}(e, s) : e \in E, s \in S\}$ on 9 environments (3 easy, 6 hard). We obtain 5 seeds * 3 envs = 15 data samples for "easy" environments, and 5 seeds * 6 envs = 30 data samples for "hard" environments, so that the "total" column has 15 + 30 = 45 data samples. In Appendix G, we increase the number of seeds from 5 to 10, so that we obtain 30 data samples for "easy" environments. In the last three columns, we aggregate the data from each policy optimization algorithm and test whether a regularizer performs significantly different from the baseline across algorithms, environments, and seeds. Since there are 4 algorithms, we obtain 15 * 4 = 60 samples for "easy", 30 * 4 = 120 for "hard", and 60 + 120 = 180 for "total". The sample size is large enough and satisfy our condition for Welch's $t$-test.

For experiments in Section 4 (e.g. Table 3), for each policy optimization algorithm, we test $\{f_{\text{reg}}(h, e, s) : h \in H, e \in E, s \in S\}$ versus $\{f_{\text{baseline}}(h, e, s) : h \in H, e \in E, s \in S\}$, where

$H$ is the set of training hyperparameters. In other words, we test whether a regularizer's performance over training hyperparameters, environments, and seeds is significantly different from that of baseline. We conducted experiments on 2 easy environments and 3 hard environments. We obtain 5 hyperparameters * 5 seeds * 2 envs = 50 data samples for "easy" environments, 5 * 5 * 3 = 75 for "hard", and 50 + 75 = 125 samples for "total". In the last three columns, we aggregate the data from each policy optimization algorithm. We obtain 50 * 4 = 200 data samples for "easy", 75 * 4 = 300 for "hard", and 200 + 300 = 500 for "total". The sample size is large enough and satisfy our condition for Welch's $t$-test.

We further plot the distribution of our $z$-score metric in the quantile-quantile (Q-Q) plots in Figure 6 and Figure 7. A Q-Q plot plots the quantiles of two distributions $X$ and $Y$ against each other, where in our case $X$ is normal. If the plot approximately follows the line $y = x$, then the two distributions have approximately the same cumulative distribution function (CDF). In our case, this means that $Y$ is approximately normal. We observe that empirically, the distribution of our performance metric is close to normal. As a result, the $t$-statistic we calculate from our samples is close to the $t$-distribution with parameter $n$, which converges to $\mathcal{N}(0, 1)$ quickly as $n$ increases.

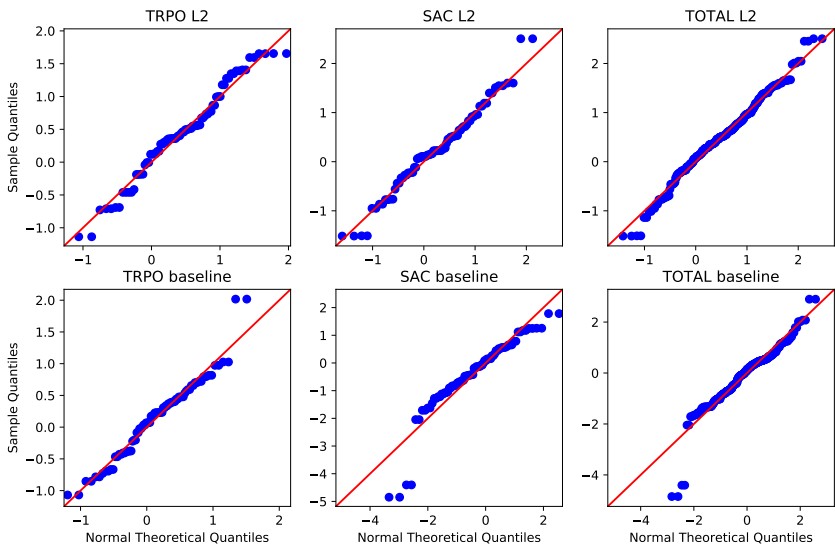

Figure 6: Quantile-Quantile (Q-Q) plot for our $z$-score metric on all environments under the default hyperparameter setting in Section 3. As an example, in the first figure, the x-axis denotes the theoretical quantile for the normal distribution with mean and standard deviation from $\{f_{\text{reg}}(e, s) : e \in E, s \in S\}$ when we apply $L_2$ regularization on TRPO. The blue points denote the actual quantiles of $\{f_{\text{reg}}(e, s) : e \in E, s \in S\}$. The red line denotes what the quantile-quantile relation looks like if $\{f_{\text{reg}}(e, s) : e \in E, s \in S\}$ were perfectly normal. We observe that the blue points are close to the red line, suggesting that the distribution of $\{f_{\text{reg}}(e, s) : e \in E, s \in S\}$ is close to a normal distribution. Similarly, we observe that $\{f_{\text{baseline}}(e, s) : e \in E, s \in S\}$ is close to normal.

We have presented the mean values for our performance metrics ($z$-scores in Table 2, 3; average ranking in Table 22, 23; scaled return in Table 26, 29). Given statistical significance, whether a regularizer improves upon baseline depends on whether the performance metric is higher than the baseline. For example, in the "Total" column and "hard" subcolumn of Table 2, the $z$-score of $L_2$ regularization is 0.58, while the $z$-score of baseline is -0.27. The $p$-value in the corresponding entry of Table 15 is 0.00. Therefore, $L_2$ regularization significantly improves over the baseline on hard tasks. Note that the $p$-value is not a standalone performance metric. It only serves as a complement to our metrics and indicates whether the performance of a regularizer differs significantly from our baseline.

In addition, we note that the Figure 5 in Henderson et al. (2018) shows that, under the same hyperparameter configuration, two sets of 5 different runs on the HalfCheetah environment can be

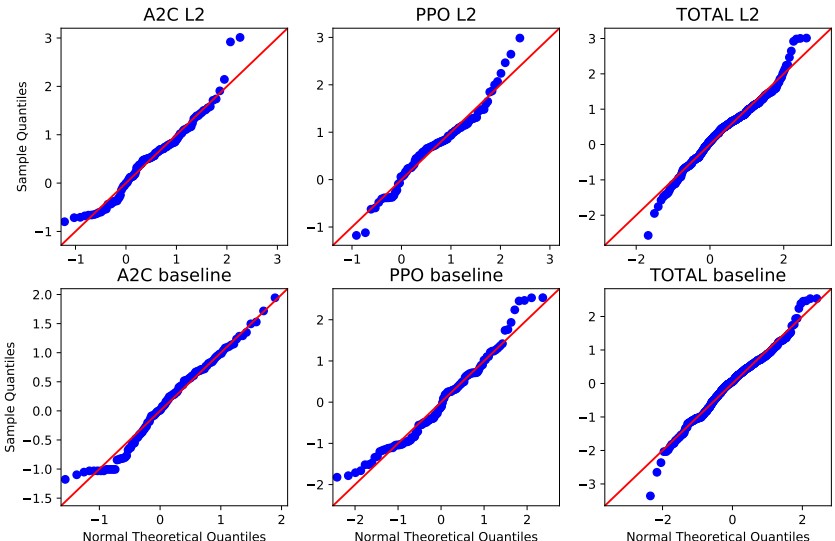

Figure 7: Quantile-Quantile (Q-Q) plot for our $z$-score metric on all environments under the 5 sampled hyperparameters in Section 4. We observe that the blue points are close to the red line, suggesting that the distribution of $\{f_{\text{reg}}(h, e, s) : h \in H, e \in E, s \in S\}$ is close to a normal distribution. Similarly, we observe that $\{f_{\text{baseline}}(h, e, s) : h \in H, e \in E, s \in S\}$ is close to normal.

significantly different from each other. We find that the unique environment property of HalfCheetah contributes to such observation. For A2C, PPO, and TRPO on HalfCheetah, there is a certain probability that the policy found is suboptimal, where the half cheetah robot runs upside-down using its head. In this case, the final return never rises above 2200. In other cases, the half cheetah robot runs using its legs, and the final return is almost always above 4000. Therefore, it is possible that in a set of 5 runs, 4 of the runs have final returns above 4000, while for another set of 5 runs, 4 of the runs have final returns below 2200. This causes a significant performance difference between the two sets of runs. However, for all other environments, the final return is approximately normally distributed with respect to seeds, instead of categorically distributed like HalfCheetah. The variance on the other environments is much smaller than that of HalfCheetah. For example, according to Table 3 of Henderson et al. (2018), PPO Walker's 95% confidence interval for the final return has a range of 800, while HalfCheetah has a range of 2200. Thus, other environments do not yield as much fluctuations as HalfCheetah. In fact, in our experiments under the default hyperparameter setting in Section 3, we do not find any regularization on any algorithm to "improve" upon HalfCheetah, according to our definition that a regularizer **"improves"** upon the baseline in Section 3. Thus, our observations do not change if we take away the HalfCheetah environment.

## L    REGULARIZATION WITH A FIXED STRENGTH

In previous sections, we tune the strength of regularization for each algorithm and environment, as described in Appendix C. Now we restrict the regularization methods to a **single** strength for each algorithm, across different environments. The results are shown in Table 32 and 33. The selected strength are presented in Table 34. We see that the $L_2$ regularization is still generally the best performing one, but SAC is an exception, where BN is better. This can be explained by the fact that in SAC, the reward scaling coefficient is different for each environment, which potentially causes the optimal $L_2$ and $L_1$ strength to vary a lot across different environments, while BN does not have a strength parameter.

Table 32: Percentage (%) of environments that, when using a regularization, "improves". For each algorithm, one single strength for each regularization is applied to all environments.

| Reg \ Alg | A2C | | | TRPO | | | PPO | | | SAC | | | TOTAL | | |
|---|---|---|---|---|---|---|---|---|---|---|---|---|---|---|---|
| | Easy | Hard | Total | Easy | Hard | Total | Easy | Hard | Total | Easy | Hard | Total | Easy | Hard | Total |
| Entropy | **33.3** | **66.7** | **55.6** | 0.0 | 33.3 | 22.2 | 0.0 | 16.7 | 11.1 | 0.0 | 16.7 | 11.1 | 8.3 | 33.3 | 25.0 |
| $L_2$ | 0.0 | 50.0 | 33.3 | 0.0 | **50.0** | **33.3** | **33.3** | **66.7** | **55.6** | 33.3 | 33.3 | 33.3 | 16.7 | **50.0** | **38.9** |
| $L_1$ | 0.0 | 33.3 | 22.2 | 0.0 | **50.0** | **33.3** | **33.3** | 50.0 | 44.4 | 33.3 | 33.3 | 33.3 | 16.7 | 41.7 | 33.3 |
| Weight clipping | 0.0 | 0.0 | 0.0 | **33.3** | 33.3 | **33.3** | **33.3** | 50.0 | 44.4 | 33.3 | 0.0 | 11.1 | 25.0 | 20.8 | 22.2 |
| Dropout | 0.0 | 0.0 | 0.0 | N/A | N/A | N/A | **33.3** | 50.0 | 44.4 | **66.7** | 16.7 | 33.3 | **33.3** | 22.2 | 25.9 |
| BatchNorm | 0.0 | 0.0 | 0.0 | 0.0 | 0.0 | 0.0 | 0.0 | 16.7 | 11.1 | 33.3 | **50.0** | **44.4** | 8.3 | 16.7 | 13.9 |

Table 33: The average z-score for different regularization methods. For each algorithm, one single strength for each regularization is applied to all environments.

| Reg \ Alg | A2C | | | TRPO | | | PPO | | | SAC | | | TOTAL | | |
|---|---|---|---|---|---|---|---|---|---|---|---|---|---|---|---|
| | Easy | Hard | Total | Easy | Hard | Total | Easy | Hard | Total | Easy | Hard | Total | Easy | Hard | Total |
| Baseline | 0.56 | 0.08 | 0.24 | **0.38** | 0.22 | 0.27 | 0.38 | -0.39 | -0.14 | 0.15 | -0.14 | -0.04 | **0.37** | -0.06 | 0.08 |
| Entropy | **1.07** | 0.86 | 0.93 | 0.25 | 0.33 | 0.3 | 0.02 | -0.25 | -0.16 | -0.03 | -0.22 | -0.16 | 0.33 | 0.18 | 0.23 |
| $L_2$ | 0.82 | **1.05** | **0.97** | 0.33 | 0.44 | 0.40 | **0.39** | **0.68** | **0.58** | -0.33 | 0.08 | -0.06 | 0.30 | **0.56** | **0.48** |
| $L_1$ | 0.39 | -0.01 | 0.12 | 0.22 | **0.63** | **0.49** | 0.32 | 0.44 | 0.40 | -0.14 | -0.15 | -0.15 | 0.20 | 0.23 | 0.22 |
| Weight Clip | -0.87 | 0.14 | -0.20 | **0.38** | 0.19 | 0.25 | **0.39** | 0.36 | 0.37 | -0.19 | -0.36 | -0.30 | -0.08 | 0.09 | 0.03 |
| Dropout | -0.96 | -1.01 | -0.99 | N/A | N/A | N/A | -0.05 | -0.19 | -0.15 | **0.61** | 0.33 | **0.43** | -0.13 | -0.29 | -0.24 |
| BatchNorm | -1.00 | -1.11 | -1.07 | -1.54 | -1.81 | -1.72 | -1.44 | -0.64 | -0.91 | -0.08 | **0.45** | 0.27 | -1.02 | -0.78 | -0.86 |

Table 34: The fixed single regularization strengths that are used in each algorithm to obtain results in Table 32 and Table 33.

| Reg \ Alg | A2C | TRPO | PPO | SAC |
|---|---|---|---|---|
| Entropy | $5e-4$ | $5e-4$ | $5e-4$ | 1.0 |
| $L_2$ | $1e-4$ | $5e-4$ | $5e-4$ | $5e-2$ |
| $L_1$ | $1e-4$ | $1e-4$ | $1e-4$ | $5e-3$ |
| Weight clipping | 0.2 | 0.2 | 0.2 | 0.3 |
| Dropout | 0.05 | 0.05 | 0.05 | 0.2 |
| BatchNorm | True | True | True | True |

# M  DDPG RESULTS

To study the effect of regularization on off-policy algorithms, besides the SAC results, we also present results on DDPG (Lillicrap et al., 2016) in Table 35. We run DDPG on 5 MuJoCo environments: Hopper, Walker, Ant, Humanoid, and HumanoidStandup. We did not run DDPG on HalfCheetah due to its large variance. We then analyze the performance through the calculation of $z$-scores, and we also perform Welch's $t$-test. Note that entropy regularization is not applicable here because DDPG's policy network outputs a deterministic action. We obtain similar observations as we did in SAC. Notably, Dropout and Batch Normalization can be useful in DDPG, as indicated by the higher average $z$-score than the baseline, which supports our hypothesis that they can be helpful on off-policy algorithms.

Table 35: The average $z$-score for each regularization method on DDPG, and $p$-value from Welch's $t$-test comparing the regularization methods to the baseline.

| Reg | $z$-score | $p$-value |
|---|---|---|
| Baseline | -0.71 | / |
| $L_2$ | 0.1 | 0.01 |
| $L_1$ | 0.44 | 0.00 |
| Weight Clip | -0.12 | 0.04 |
| Dropout | 0.47 | 0.00 |
| BatchNorm | -0.17 | 0.05 |

# N REGULARIZING WITH $L_2$ AND ENTROPY

We also investigate the effect of combining $L_2$ regularization with entropy regularization, given that both cases of applying one of them alone yield performance improvement. We take the optimal strength of $L_2$ regularization and entropy regularization together and compare with applying $L_2$ regularization or entropy regularization alone. From Figure 8, we find that the performance increases for PPO HumanoidStandup, approximately stays the same for TRPO Ant, and decreases for A2C HumanoidStandup. Thus, the regularization benefits are not always addable. This phenomenon is possibly caused by the fact that the algorithms already achieve good performance using only $L_2$ regularization or entropy regularization, and further performance improvement is restrained by the intrinsic capabilities of algorithms.

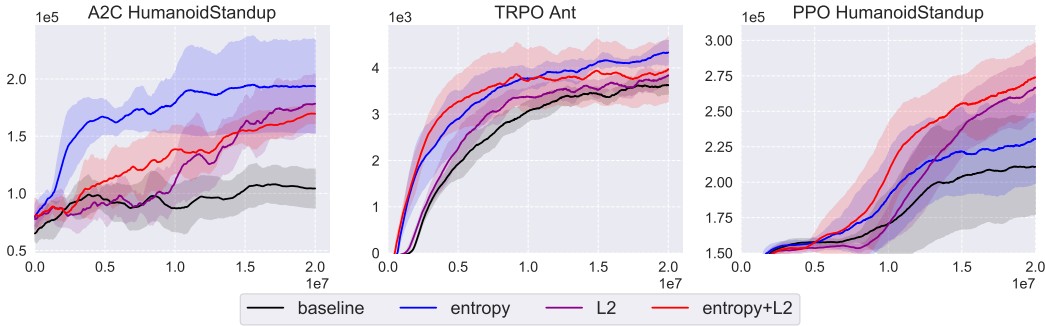

Figure 8: The effect of combining $L_2$ regularization with entropy regularization. For PPO HumanoidStandup, we use the third randomly sampled hyperparameter setting. For A2C HumanoidStandup and TRPO Ant, we use the baseline as in Section 3.

## O $L_2$ REGULARIZATION VS. FIXED WEIGHT DECAY (ADAMW)

For the Adam optimizer (Kingma & Ba, 2015), "fixed weight decay" (AdamW in Loshchilov & Hutter (2019)) differs from $L_2$ regularization in that the gradient of $\frac{1}{2}\lambda||\theta||^2$ is not computed with the gradient of the original loss, but the weight is "decayed" finally with the gradient update. For Adam these two procedures are very different (see Loshchilov & Hutter (2019) for more details). In this section, we compare the effect of adding $L_2$ regularization with that of using AdamW, with PPO on Humanoid and HumanoidStandup. The result is shown in Figure 9. Similar to $L_2$, we briefly tune the strength of weight decay in AdamW and the optimal one is used. We find that while both $L_2$ regularization and AdamW can significantly improve the performance over baseline, the performance of AdamW tends to be slightly lower than the performance of $L_2$ regularization.

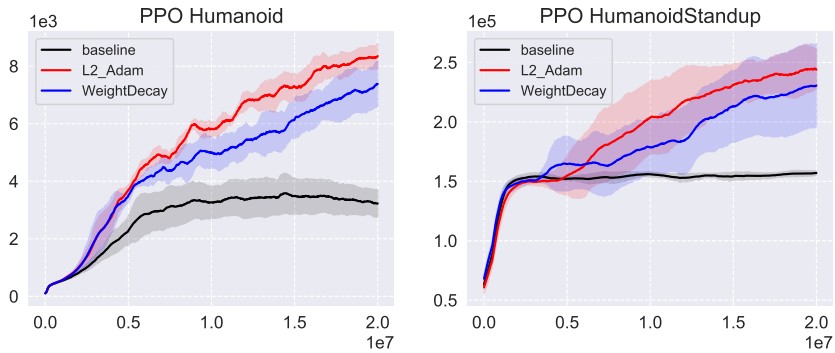

Figure 9: Comparison between $L_2$ regularization and weight decay. For PPO Humanoid and HumanoidStandup, we use the third randomly sampled hyperparameter setting.

# P   ADDITIONAL TRAINING CURVES (DEFAULT HYPERPARAMETERS)

Figure 10: Return vs. timesteps, for four algorithms (columns) and five environments (rows).

As a complement with Figure 1 in Section 3, we plot the training curves of the other five environments in Figure 10.

# Q  TRAINING CURVES FOR HYPERPARAMETER EXPERIMENTS

In this section, we plot the full training curves of the experiments in Section 4 with five sampled hyperparameter settings for each algorithm in Figure 11 to 14. The strength of each regularization is tuned according to the range in Appendix C.

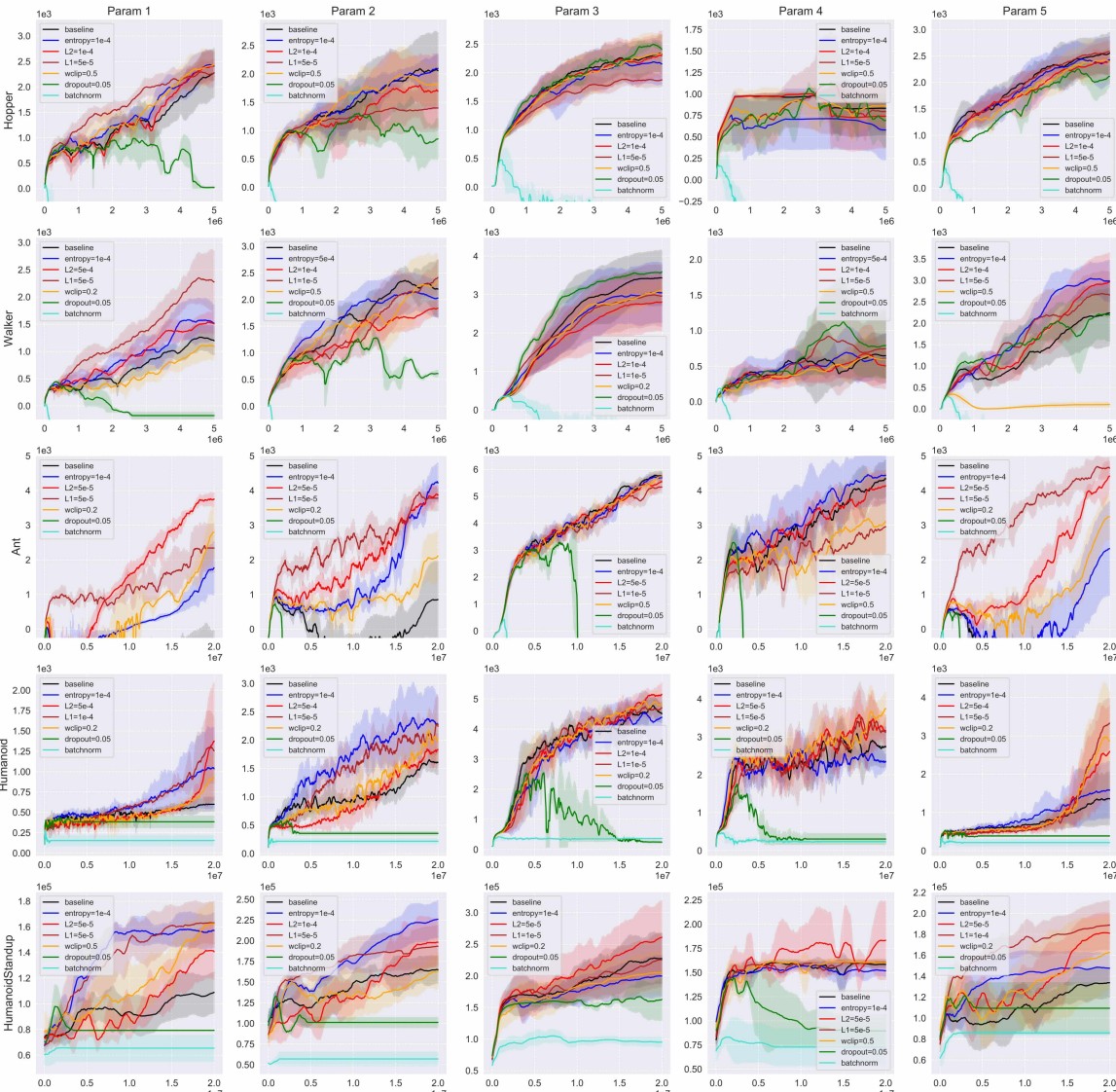

Figure 11: Training curves of A2C regularizations under five randomly sampled hyperparameters.

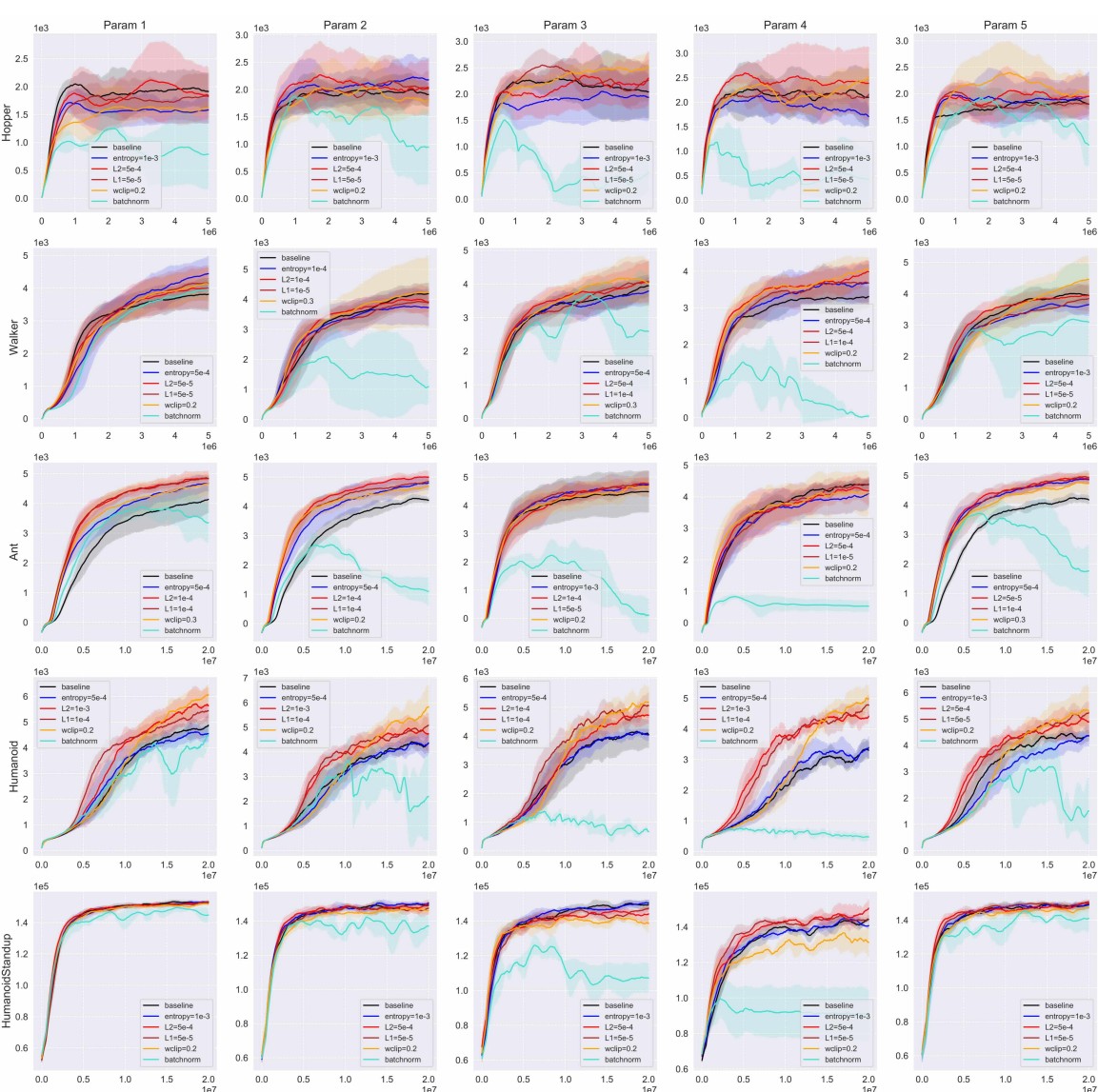

Figure 12: Training curves of TRPO regularizations under five randomly sampled hyperparameters.

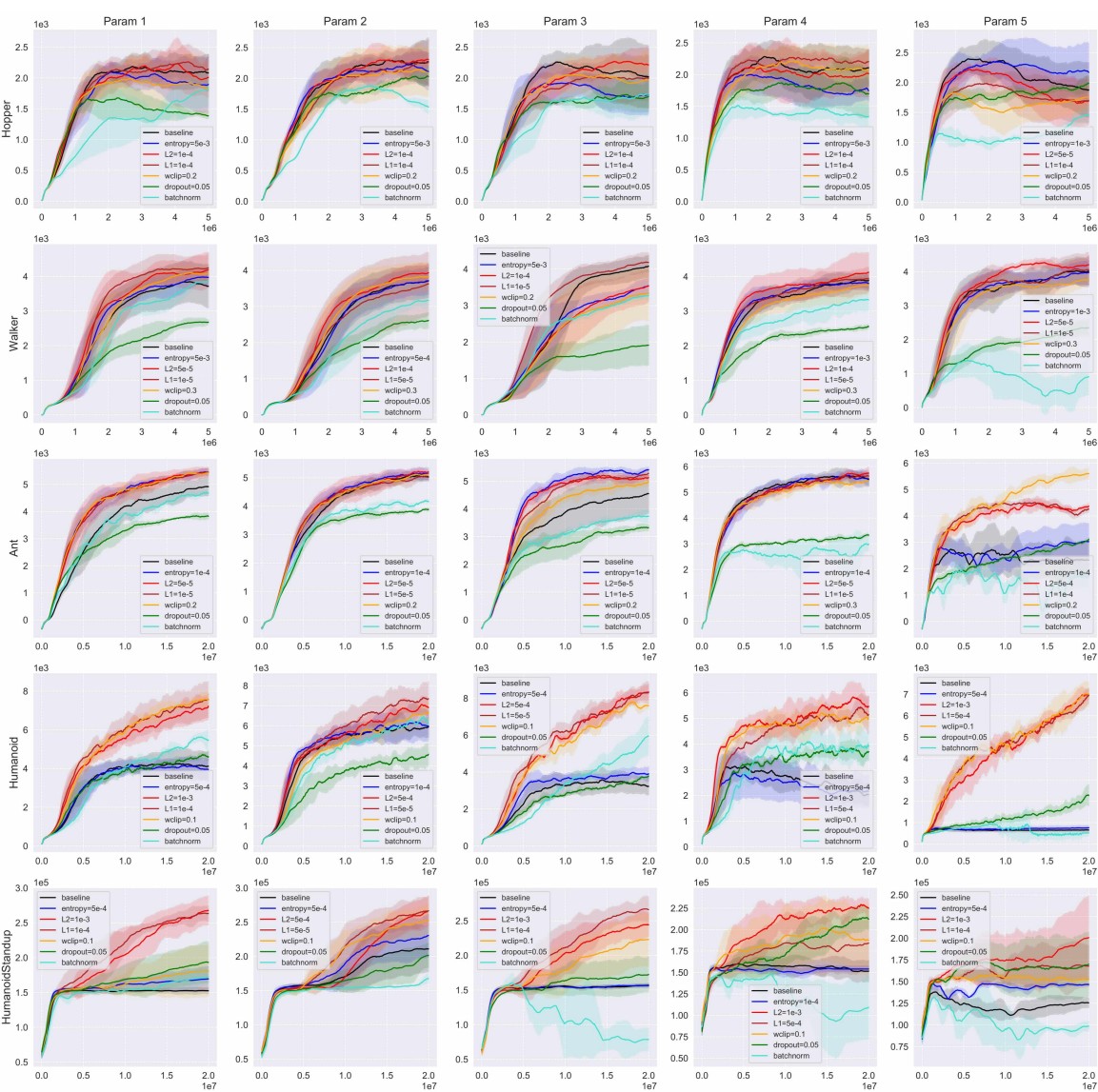

Figure 13: Training curves of PPO regularizations under five randomly sampled hyperparameters.

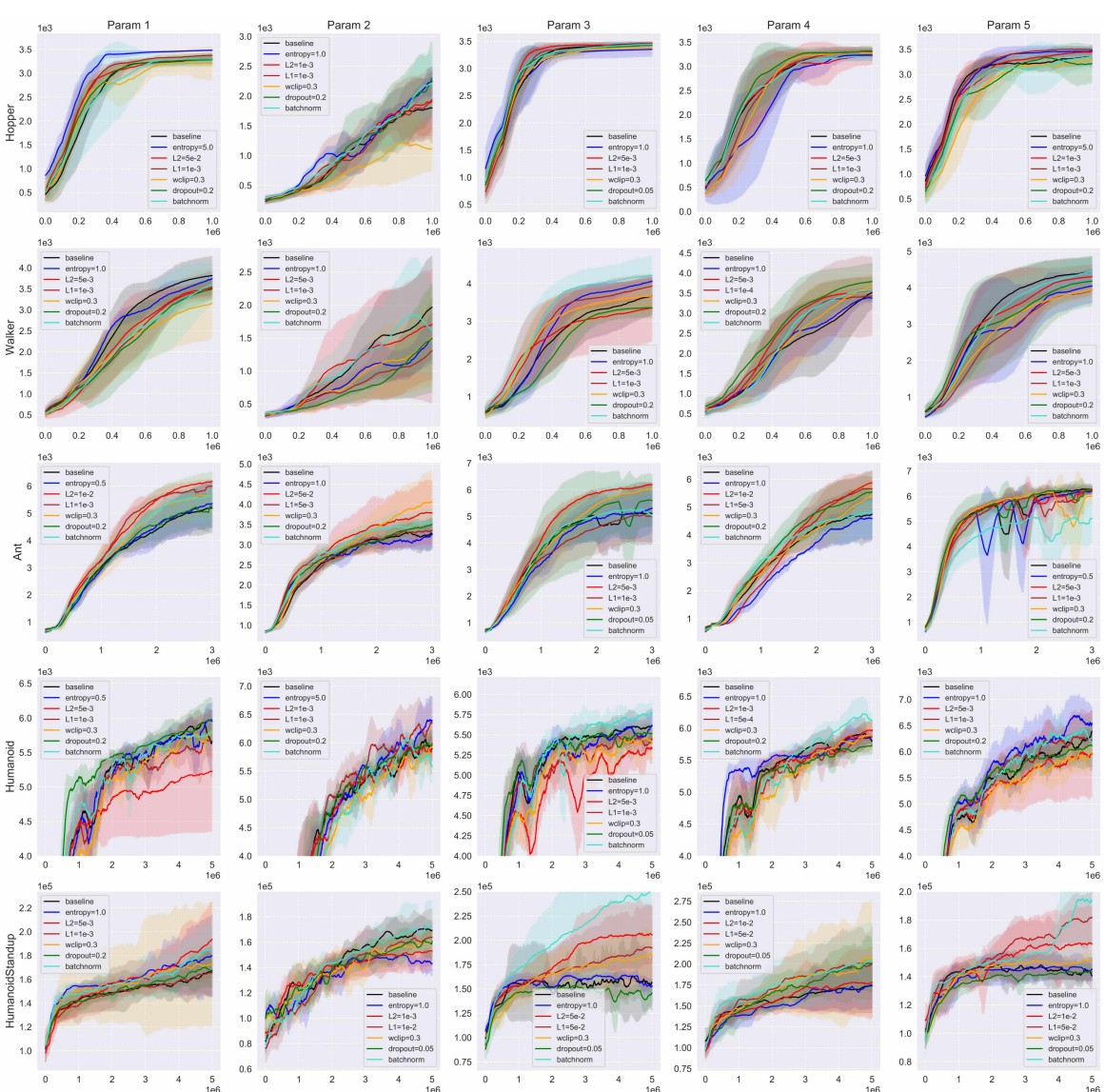

Figure 14: Training curves of SAC regularizations under five randomly sampled hyperparameters.

# R   TRAINING CURVES FOR POLICY VS. VALUE EXPERIMENTS

We plot the training curves with our study in Section 5 on policy and value network regularizations in Figure 15-18.

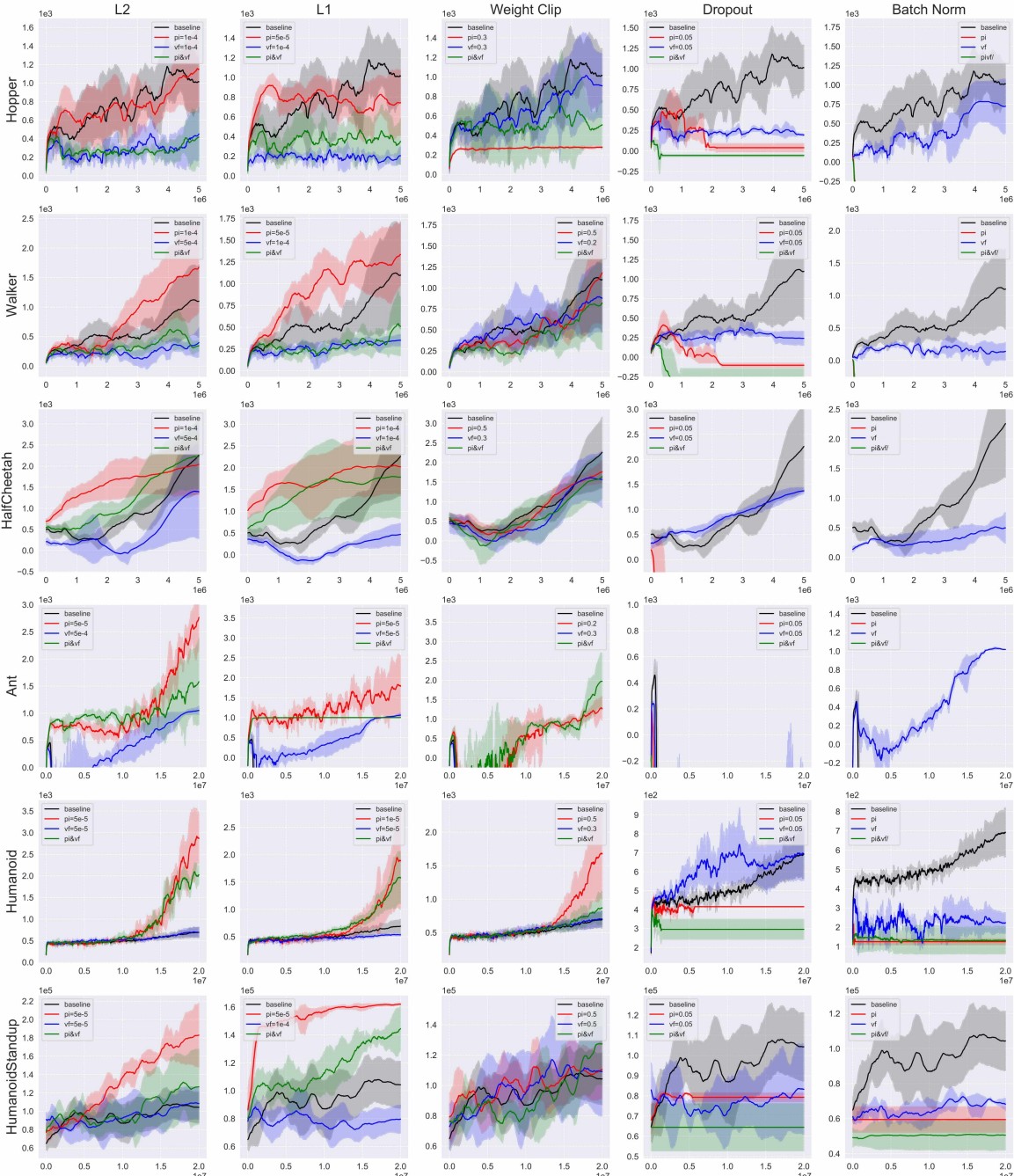

Figure 15: The interaction between policy and value network regularization for A2C. The optimal policy regularization and value regularization strengths are listed in the legends. Results of regularizing both policy and value networks are obtained by combining the optimal policy and value regularization strengths.

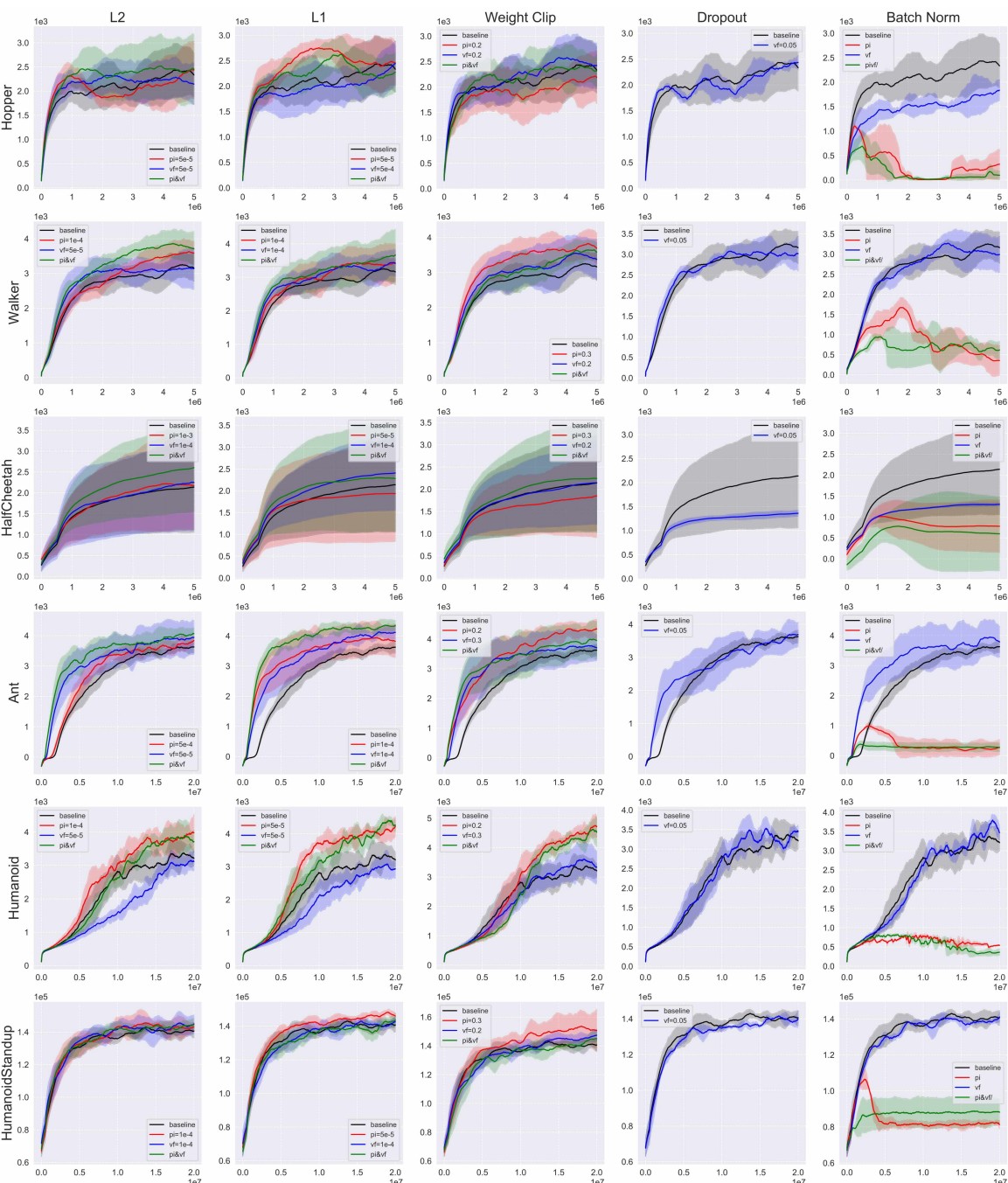

Figure 16: The interaction between policy and value network regularization for TRPO.

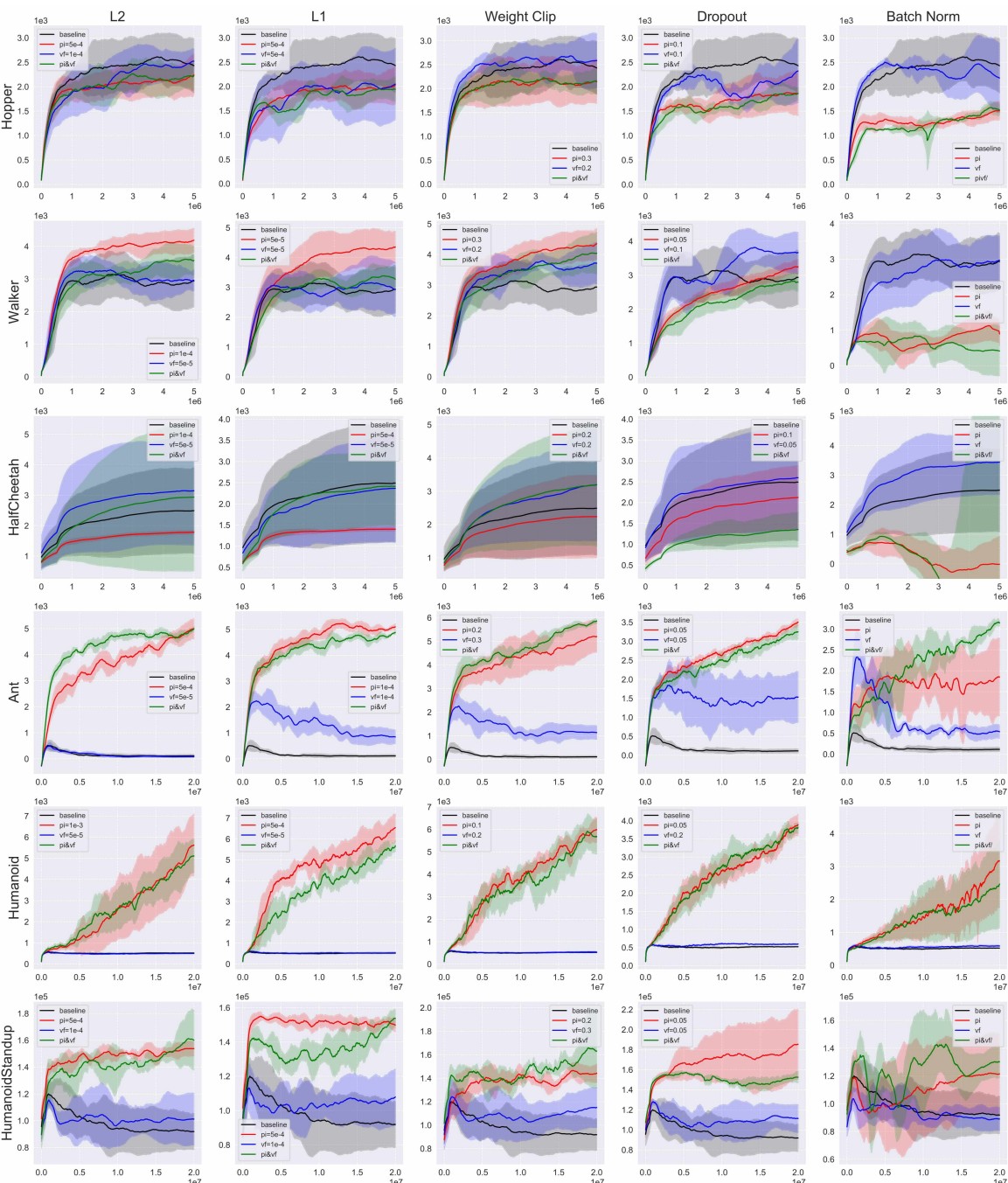

Figure 17: The interaction between policy and value network regularization for PPO.

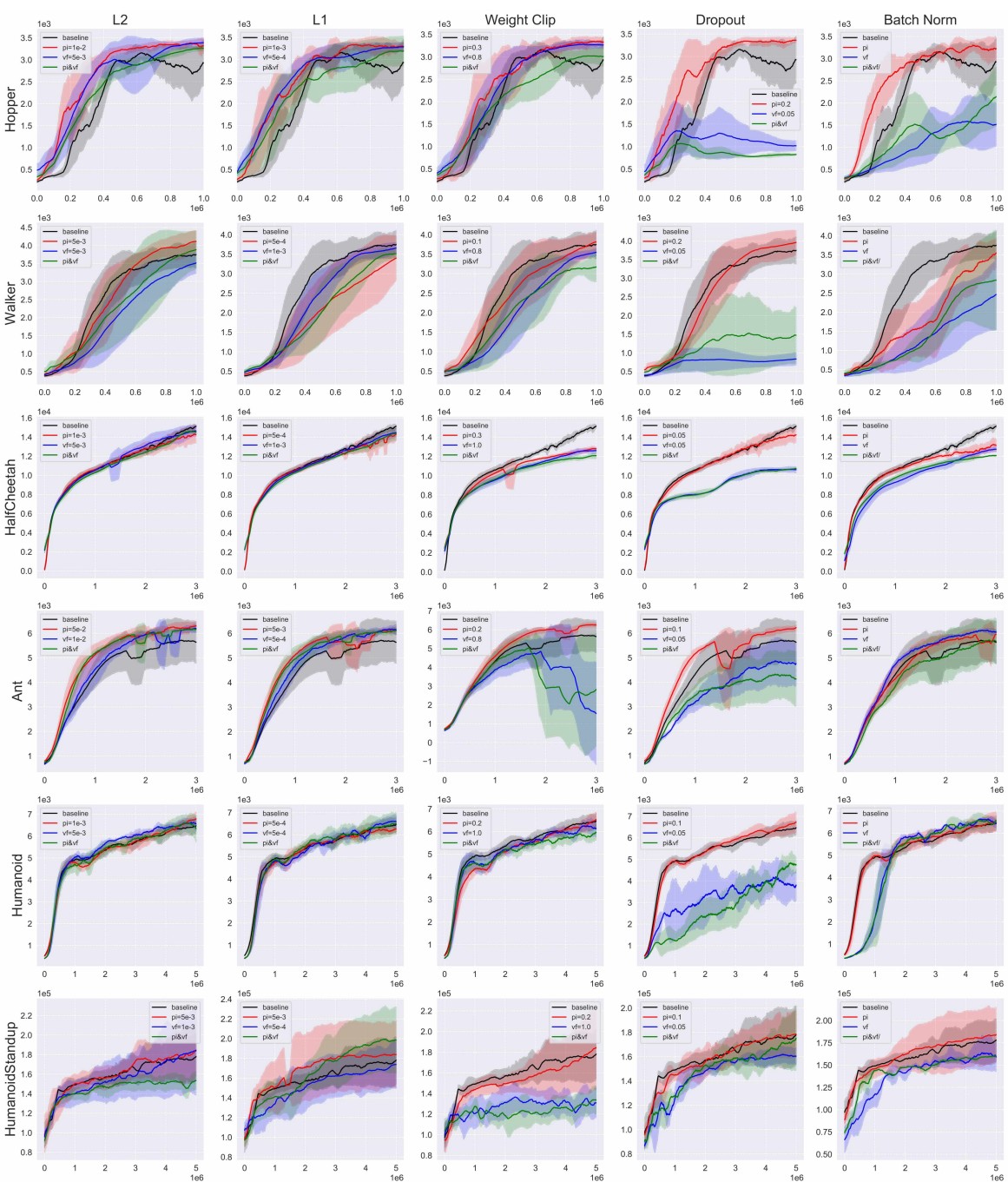

Figure 18: The interaction between policy and value network regularization for SAC.

# S    ATARI EXPERIMENTS

We present results on 5 randomly sampled Atari environments (Asteroids, Pacman, Qbert, Roadrunner, Riverraid) in Table 36. Note that SAC not applicable here because it requires the environment to have continuous action space, while Atari environments have discrete action space. We find that $L_2$ regularization can still significantly improve over the baseline, while $L_1$ and weight clipping are slightly less effective. Interestingly, while BN still significantly harms performance on on-policy environments (A2C, TRPO, PPO), dropout can significantly outperform the baseline. We also observe that, different from continuous control tasks, entropy regularization can improve a lot on baseline, perhaps due to the action space being discrete.

Table 36: Average $z$-scores on Atari envs with $p$-values testing regularizers against baseline.

| Reg \ Alg | A2C | TRPO | PPO | TOTAL | $p$-value |
|---|---|---|---|---|---|
| Baseline | 0.10 | 0.07 | 0.03 | 0.07 | N/A |
| Entropy | 0.59 | 0.24 | 0.65 | 0.49 | 0.00 |
| $L_2$ | 0.23 | 0.49 | 0.39 | 0.37 | 0.01 |
| $L_1$ | 0.01 | 0.15 | 0.15 | 0.10 | 0.73 |
| Weight Clip | 0.14 | -0.16 | 0.09 | 0.03 | 0.74 |
| Dropout | 0.43 | N/A | 0.55 | 0.49 | 0.00 |
| BatchNorm | -1.51 | -0.79 | -1.86 | -1.39 | 0.00 |

