# OpenReview forum: "Regularization Matters in Policy Optimization - An Empirical Study on Continuous Control"
_ICLR.cc/2021/Conference — ICLR 2021 Spotlight_

### Official Review · AnonReviewer2 · 2020-10-24
**Statistical significance is an issue**

**Rating:** 7
**Confidence:** 4

**Review:**

**Pros**

The paper investigates a worthwhile question. Given the prevalence and success of regularization in deep learning, it is curious why we haven't consistently observed the same in deep RL.

**Cons**

My main concern with the work, especially since it is an empirical study, concerned the lack of statistical significance in the results. Here are the problems I found.

1. Welch's t-test requires normality assumption, and I'm not sure why returns would be normally distributed; in any case, no verification of this assumption was attempted.
2. Even if the normality issue is ignored, low p-values in the t-test suggests that the means are different, but doesn't necessarily say anything about the direction or magnitude of difference. It could be that regularisation worsens performance, for all we know.
3. The learning curves are uninformative for determining the direction and magnitude of any performance difference, given the high variance of RL and the number of seeds used (5 seeds). Indeed, in the Henderson et al. work cited, Figure 5 shows how a small number of random seeds can be misleading (they even show that for the same algorithm and different seeds, a t-test can give a small p-value).
4. Similar concerns apply for ranking metrics, given the small sample size.

**Summary**

Based on the concerns with statistical significance I have highlighted above, I recommend rejection of this paper. The results presented in the paper are certainly suggestive, but do not I think meet the level of rigour required at ICLR. I do very much like the motivation of the study, and would have liked to have seen results of greater statistical significance.

I recognize that the results presented in the text already took 57 days on a substantial computational setup, but it worries me that we are using massive computational power to little avail if we cannot produce reliable results. It would perhaps be better to focus on a smaller subset of algorithms and environments in order to guarantee reliable results.

** Edit after author response **
I feel that the authors have sufficiently addressed the statistical significance concerns I had in the comments below. I now recommend acceptance because I think the work provides strong evidence that regularization is beneficial in continuous control. Although, as other reviewers have pointed out, further analysis of why regularization is beneficial--especially from a theoretical standpoint--would be helpful, I think the empirical contribution of the paper still stands.

---

> ### Author Response · Authors · 2020-11-20
> **Thanks for your review; addressing concerns below [3/3]**
>
> (Continued for 3a) **In our updated version, we increase the number of seeds from 5 to 10** and present results for the six MuJoCo environments (easy: Hopper, Walker, HalfCheetah; hard: Ant, Humanoid, HumanoidStandup) in Appendix G to further verify our observations. Due to the large computation cost required, we do not include the three hard Roboschool environments.  We report $z$-scores, and we also conduct Welch’s $t$-test (for each algorithm, “easy”: 3 envs * 10 seeds = 30 samples; “hard”: 3 * 10 = 30 samples; “total”: 30 + 30 = 60 samples; for the last three columns on statistics over all algorithms, “easy”: 30 * 4 = 120; “hard”: 30 * 4 = 120; “total”: 120 + 120 = 240; thus the sample size is large enough for $t$-test).
>
> **We find that our observations in Section 3 still hold, i.e. our observations from 5 seeds per environment are consistent with those from 10 seeds per environment.** For example, $L_2$ tops the average $z$-score most often, and by large margin in total; entropy regularization is best used with A2C; Dropout and BN are only useful in the off-policy SAC algorithm; the improvement over baseline is larger on hard tasks.
>
> >3b. In the Henderson et al. work cited, Figure 5 shows how a small number of random seeds can be misleading (they even show that for the same algorithm and different seeds, a t-test can give a small p-value).
>
> **The Figure 5 of Henderson et al. [4] tests a set of 5 random seeds vs another set of 5 seeds on the single environment of HalfCheetah. We would like to clarify that our $t$-test does not test whether a regularizer significantly improves over the baseline under one single environment, but under all environments. Thus, our sample size satisfies the condition for statistical tests. More details are presented in our response to the first concern.**
>
> **We further discuss the HalfCheetah environment used in Figure 5 in Henderson et al.**, which shows that, under the same hyperparameter configuration, two sets of 5 different runs on the HalfCheetah environment can be significantly different from each other. **We find that the unique environment property of HalfCheetah contributes to such observation.** For A2C, PPO, and TRPO on HalfCheetah, there is a certain probability that the policy found is suboptimal, where the half cheetah robot runs upside-down using its head. In this case, the final return never rises above 2200. In other cases, the half cheetah robot runs using its legs, and the final return is almost always above 4000. Therefore, it is possible that in a set of 5 runs, 4 of the runs have final returns above 4000, while for another set of 5 runs, 4 of the runs have final returns below 2200. This causes a significant performance difference between the two sets of runs. However, for all other environments, the final return is approximately normally distributed with respect to seeds, instead of categorically distributed like HalfCheetah. The variance on the other environments is much smaller than that of HalfCheetah. For example, according to Table 3 of Henderson et al., PPO Walker's 95% confidence interval for the final return has a range of 800, while HalfCheetah has a range of 2200. **Empirically, we found that other environments do not yield as much fluctuations as HalfCheetah.**
>
>
> >4. Similar concerns apply for ranking metrics, given the small sample size.
>
>
>
> **Since we test on all environments instead of on a single environment, the sample size is large enough to satisfy the condition for statistical significance tests.** We have conducted $t$-test on the ranking metrics in Table 24 and 25.
>
> We would like to note that **the ranking metric is only one of the metrics we evaluate in our paper**. We have also evaluated improvement percentage metric (Table 1,4,19), $z$-score (Table 2, 3), and scaled returns (Table 26, 29). We believe evaluating under a variety of metrics make our conclusions more reliable.
>
>
> >5. Based on the concerns with statistical significance I have highlighted above ... cannot produce reliable results
>
> To summarize our response,
> - **We believe that our results are reliable, because we evaluate our results on multiple performance metrics, and the condition for significance testing is satisfied.**
> - **To further address your concerns, in our updated version, we report the results on MuJoCo when we increase the number of training seeds from 5 to 10, and we find that our observations are consistent.**
>
> -----
>
> We hope our response could address your concern, and we thank you again for the helpful comments. We are glad to discuss further comments and suggestions.
>
> [1] Location Test. https://en.wikipedia.org/wiki/Location_test
> [2] Statistics/Testing Data/t-tests. https://en.wikibooks.org/wiki/Statistics/Testing_Data/t-tests
> [3] Slutsky's Theorem. https://en.wikipedia.org/wiki/Slutsky%27s_theorem
> [4] Henderson et al., Deep Reinforcement Learning that Matters

---

> > ### Comment · AnonReviewer2 · 2020-11-21
> > **additional concerns**
> >
> > In reading this detailed response, I had a side thought about the greater validity of the experiments. If I understand correctly, the hyperparameters were not tuned for the MuJoCo tasks. For the hyperparameters that were tuned in Roboschool, it seems like the architecture parameters were not tuned. It seems to me that that implies the conclusions of the study are limited to "the algorithms with the hyperparameters given" (e.g., PPO with its parameters), rather than about the algorithms themselves. Presumably, PPO with its given hyperparameters and architecture is not going to be valid for every single task, and regularization may interact in unforeseen ways with the architecture (see https://arxiv.org/abs/2010.09163 as an example of studying architectural choices in RL, fig 2 of https://arxiv.org/pdf/1709.06560.pdf). I understand that this was looked into to some extent in section 4, but it seems possible to observe different results as the changing of different hyperparameters, architecture included, interact with each other.
> >
> > Any thoughts on this (from any of the other reviewers too)?

---

> > > ### Author Response · Authors · 2020-11-22
> > > **Response and clarifications on our hyperparameter robustness experiments**
> > >
> > > Thank you for your reply and we would like to address your additional concern below.
> > >
> > > In Section 3, it's indeed the case that we did not tune the hyperparameters for the MuJoCo tasks, and that we briefly tuned the hyperparameters for the Roboschool tasks, because the codebases we adopted (OpenAI and SAC official release) do provide default hyperparameters for MuJoCo tasks, but not for Roboschool tasks.
> > >
> > > We are aware that RL experiment results can be sensitive to hyperparameters changes (Henderson et al.), and any observations drawn can be vulnerable to such variations. Therefore, **to confirm our findings hold under different hyperparameters, in Section 4. we sample multiple hyperparameter settings for each algorithm and conduct experiments.** The sampled hyperparameters are listed in Appendix E, and we also plotted training curves from all hyperparameter settings in Appendix Q. We observe **our findings with Secton 3 still hold.** **Therefore, our findings are not only limited to a specific set of hyperparameters for each algorithm.** We agree it is possible to observe different results as the changing of different hyperparameters, architecture included, interact with each other, but we would like to clarify that:
> > >
> > >
> > > 1. As with most RL/ML experiments, returns/performance typically change with different hyperparameters, but as we show in Table 3 (z-score) and Table 16 (significance test), **on average, certain regularizers ($L_2$, $L_1$) still outperform the baseline with statistical significance**. The improvement percentage in Table 19, where we consider each (hyperparameter, task) pair to be a single case, shows that **the same observations as in Table 1 hold in most cases**. Thus, the outperformance of these regulariziers is not due to a small subset of specific hyperparameter settings.
> > >
> > > 2. **In Figure 2, where we plot the change of return with a single varying hyperparameter** (such as ***architecture width/depth*, learning rate, and rollout timesteps**), we also note that **in most cases, regularizers like $L_2$ and $L_1$ outperform the baseline. With certain regularizers, such as $L_2$ and $L_1$, we can see that the returns fluctate much less over different hyperparamter variations compared to the baseline.** From another perspective, this means sometimes with suboptimal hyperparameters, training with proper regularization can boost the return to the same level as baseline with better hyperparameters. This demonstrate using proper regularization can make the algorithm more robust against hyperparemeter changes and ease their tuning.
> > >
> > >
> > > In summary, we believe our experiments in Section 4 demonstrated that our findings are not due to any specific hyperparameter setting, but are robust against hyperparameter variations.

---

> ### Author Response · Authors · 2020-11-20
> **Thanks for your review; addressing concerns below [2/3]**
>
> (Continued for 1) **We now show that our sample size is large enough to apply the above theorem.** For each algorithm and regularizer, we calculate the average $z$-score, the average ranking, and the average scaled return over a set of environments and all seeds. We then test whether the performance of a regularizer is significantly different from that of baseline. We take the average $z$-score metric as an example. Let $E$ be the set of environments with uniform distribution over the environments, and let $S$ be the set of seeds. For $e\in E$ and $s \in S$, let $f_{\text{reg}}(e, s)$ denote the $z$-score of a certain regularizer under an environment $e$ and seed $s$, and let $f_{\text{baseline}}(e, s)$ denote the $z$-score under the baseline. We use Welch's $t$-test to test whether $\mu_{\text{reg}} \neq \mu_{\text{baseline}}$, given unknown $\sigma_{\text{reg}},\sigma_{\text{baseline}}$, on a policy optimization algorithm.
>
> For experiments in Section 3 (e.g. Table 2), for each policy optimization algorithm, we test the distribution of $\\{f_{\text{reg}}(e, s): e\in E, s\in S\\}$ versus $\\{f_{\text{baseline}}(e, s): e\in E, s\in S\\}$ on 9 environments (3 easy, 6 hard). We obtain 5 seeds * 3 envs = 15 data samples for easy environments, and 5 seeds * 6 envs = 30 data samples for hard environments, so that the "total" column has 15 + 30 = 45 data samples. (In Appendix G of our updated version, we increase the number of seeds from 5 to 10, so that we obtain 30 data samples for "easy" environments). In the last three columns, we aggregate the data from each policy optimization algorithm and test whether a regularizer performs significantly different from the baseline across algorithms, environments, and seeds. Since there are 4 algorithms, we obtain 15 * 4 = 60 samples for "easy", 30 * 4 = 120 for "hard", and 60 + 120 = 180 for "total". The sample size is large enough and satisfy our condition for Welch's $t$-test.
>
> For experiments in Section 4 (e.g. Table 3), for each policy optimization algorithm, we test the distribution of $\\{f_{\text{reg}}(h, e, s): h\in H, e\in E, s\in S\\}$ versus $\\{f_{\text{baseline}}(h, e, s): h\in H, e\in E, s\in S\\}$, where $H$ is the set of training hyperparameters. In other words, we test whether a regularizer's performance over training hyperparameters, environments, and seeds is significantly different from that of baseline. We conducted experiments on 2 easy environments and 3 hard environments. We obtain 5 hyperparameters * 5 seeds * 2 envs = 50 data samples for "easy" environments, 5 * 5 * 3 = 75 for "hard", and 50 + 75 = 125 samples for "total". In the last three columns, we aggregate the data from each policy optimization algorithm. We obtain 50 * 4 = 200 data samples for "easy", 75 * 4 = 300 for "hard", and 200 + 300 = 500 for "total". The sample size is large enough and satisfy our condition for Welch's $t$-test.
>
>
>
> >2. Low p-values in the t-test suggests that the means are different, but doesn't necessarily say anything about the direction or magnitude of difference; it could be that regularisation worsens performance.
>
> **We would like to clarify that we have presented the mean values for our performance metrics (average $z$-scores in Table 2,3; average ranking in Table 22,23; average scaled return in Table 26, 29). The directions and magnitudes of difference can be read from the mean values.** For example, in the "Total" column and "hard" subcolumn of Table 2, the $z$-score of $L_2$ regularization is 0.58, while the $z$-score of baseline is -0.27. The $p$-value in the corresponding entry of Table 15 is 0.00. Therefore, $L_2$ regularization significantly improves over the baseline on hard tasks. **Note that the $p$-value is not a standalone performance metric.** It only serves as a complement to our metrics and indicates whether the performance of a regularizer differs significantly from our baseline. In the updated version, we have added this clarification in Appendix K.
>
>
>
> >3a. The learning curves are uninformative for determining the direction and magnitude of any performance difference, given the high variance of RL and the number of seeds used (5 seeds).
>
> **We would like to note that the learning curves only provide qualitative results to complement with our quantitative metrics to better view how the returns are changing throughout the training process. To determine the exact magnitude and direction of performance difference, we have calculated multiple quantitative metrics (improvement percentage, $z$-scores, scaled return, average ranking) and provided statistical significance tests for more rigorous analysis.**

---

> > ### Comment · AnonReviewer2 · 2020-11-21
> > **response**
> >
> > Thank you for the response; I feel that points 2 and 3 have been clarified for me now.

---

> > > ### Author Response · Authors · 2020-11-22
> > > **Thanks for your reply**
> > >
> > > Thanks for your reply and we are glad our answers helped address your concerns.

---

> ### Author Response · Authors · 2020-11-20
> **Thanks for your review; addressing concerns below [1/3]**
>
> We sincerely thank you for your constructive comments. We are encouraged that you found our research problem worthwhile and liked the motivation of our study. We would like to address the comments and questions below, and we have updated our submission accordingly.
>
> >1. Welch's t-test requires normality assumption, and I'm not sure why returns would be normally distributed; in any case, no verification of this assumption was attempted.
>
> **We would like to clarify that (1) when the sample size is large enough $(n\ge 30)$ , the normality assumption for the sampling distribution is not needed (See the Table "Parametric and nonparametric location tests", 2 groups, independent, $N\ge 30$, in [1] https://en.wikipedia.org/wiki/Location_test); (2) since we test on the entire set of environments instead of on a single environment, our sample size is large enough to satisfy the condition of Welch's $t$-test.** We provide our justifications in detail below. In the updated version, we have also added our justifications in Appendix K.
>
> Consider two distributions with mean and variance pairs $(\mu_1, \sigma_1^2)$ and $(\mu_2, \sigma_2^2)$, respectively, where **neither distribution needs to be normal**, and the mean and variances are unknown. Let $H_0: \mu_1=\mu_2$ be the null hypothesis, and $H_1: \mu_1\neq \mu_2$ be the alternate hypothesis. Let $(X_1,X_2, \dots, X_n)$ and $(Y_1,Y_2, \dots, Y_n)$ be independent samples from the two distributions. Then, under the null hypothesis, the $t$ statistic from Welch's $t$-test converges in distribution to $\mathcal{N}(0, 1)$ as $n\to \infty$. We formalize the above statement below.
>
> **Theorem.** Consider two distributions with mean and variance pairs $(\mu_1, \sigma_1^2)$ and $(\mu_2, \sigma_2^2)$, where the mean and variances are unknown. Define $H_0: \mu_1=\mu_2$ and $H_1: \mu_1\neq \mu_2$. Let $(X_1,X_2, \dots, X_n)$ and $(Y_1,Y_2, \dots, Y_n)$ be independent samples from the two distributions. Then, under $H_0$, the $t$ statistic from Welch's $t$-test converges in distribution to the standard normal distribution as $n\to \infty$. That is, $t_{n} = \\frac{\\sqrt{n} ( \\overline{X_n} - \overline{Y_n})}{\sqrt{S_{X,n} ^2 + S_{Y,n} ^2}} \overset{d}{\longrightarrow} \mathcal{N} (0,1)$, where $\overline{X_n}, \overline{Y_n}$ are the sample means for $(X_1,X_2, \dots, X_n)$ and $(Y_1,Y_2, \dots, Y_n)$; $S_{X,n} ^2$ and $S_{Y,n} ^2$ are the sample variances.
>
> Proof.
>
> We have $S_{X,n} ^2 \overset{p}{\longrightarrow} \sigma_1 ^2$ and $S_{Y,n} ^2 \overset{p}{\longrightarrow} \sigma_2 ^2$. Then due to independence, $(S_{X,n} ^2, S_{Y,n} ^2) \overset{p}{\longrightarrow} (\sigma_1 ^2, \sigma_2 ^2)$. By the continuous mapping theorem, $\sqrt{S_{X,n} ^2 + S_{Y,n} ^2} \overset{p}{\longrightarrow} \sqrt{\sigma_1 ^2 + \sigma_2 ^2 }$.
>
> By the definition of significance tests, the rejection / acceptance region of $t_n$ is based on the null hypothesis $\mu_1=\mu_2$. Under the null hypothesis, according to the Central Limit Theorem, $\sqrt{n} (\overline{X}_n - \mu_1 )
>  \overset{d}{\longrightarrow} \mathcal{N} (0, \sigma_1 ^2)$, $\sqrt{n} (\overline{Y}_n - \mu_1 ) \overset{d}{\longrightarrow} \mathcal{N} (0, \sigma_2 ^2)$.
>
> Then due to independence, $(\sqrt{n} (\overline{X}_n - \mu_1 ), \sqrt{n} (\overline{Y}_n - \mu_1 )) \overset{d}{\longrightarrow} (\mathcal{N} (0, \sigma_1 ^2), \mathcal{N} (0, \sigma_2 ^2))$. By the continuous mapping theorem, $\sqrt{n} (\overline{X}_n - \overline{Y}_n ) \overset{d}{\longrightarrow} \mathcal{N} (0, \sigma_1 ^2 + \sigma_2 ^2 )$.
>
>  By Slutsky's theorem [3], $\frac{\sqrt{n}(\overline{X_n} - \overline{Y_n})}{\sqrt{S_{X,n} ^2 + S_{Y,n} ^2}}\overset{d} {\longrightarrow} \mathcal{N} (0,1)$.   $\square$
>
> **Therefore, if the sample size is large (i.e. $n\ge 30$), we do not need the sampling distribution to be normal to apply Welch's $t$-test [1]. Since $t_n$ converges in distribution to $\mathcal{N}(0,1)$, we can use our $t$-statistic to obtain the $p$-value the same way as from the $z$-test** (i.e. the $p$-value equals $2\cdot \Phi(-|t|)$, where $\Phi$ is the CDF of the standard normal distribution) [2]. Also, the $t$-test can be applied when $n$ grows much larger than 30 [2].

---

> > ### Comment · AnonReviewer2 · 2020-11-21
> > **response to normality**
> >
> > Thank you for the detailed response, I really appreciate it!
> >
> > I agree with the derivation above, but I'm a little caught up on the reasoning. It's true that the statistic you have is asymptotically normal, but I'm unclear as to how that justifies applying the Welch t-test if the sample-size is large since normality is still an assumption (https://en.wikipedia.org/wiki/Welch%27s_t-test). I would certainly agree that having a large sample size would make Welch's t-test a more suitable choice than if there were a small sample size, but the convergence rate of the CLT can vary depending on the moments of the underlying distribution (e.g., Berry-Essen). I think my concerns in this regard would be allayed if there were some normality check (even visual) of the statistic.

---

> > > ### Author Response · Authors · 2020-11-22
> > > **Response to normality check (Q-Q plot in Appendix K)**
> > >
> > > Thank you for your reply! We address your concern further below:
> > >
> > > The statistic $t_n$ above is equal to the Welch's $t$-test statistic (given that the sample sizes from the two distributions are the same). Regardless of whether the two distributions are normal, $t_n$ is asymptotically normal, which means that when the sample size is large, under the null hypothesis, we can approximate the distribution of $t_n$ with the standard normal distribution. **Therefore, when $n$ is large, we no longer require the assumption that the two distributions are normal, and we can use $\mathcal{N}(0,1)$ to approximate the distribution of $t_n$. This means that when $n$ is large, we can apply Welch's $t$-test. The "advantages and limitations" section in [1] (https://en.wikipedia.org/wiki/Welch%27s_t-test) mentions that "Welch's t-test remains robust for skewed distributions and large sample sizes". Also, the Table "Parametric and nonparametric location tests", 2 groups, independent, $N\ge 30$ in [2] (https://en.wikipedia.org/wiki/Location_test) shows that when our $n \ge 30$, it does not require that the distributions are normal.**
> > >
> > >
> > >
> > > **To address your concern about the convergence rate of CLT, in our newly updated version, we plot the quantile-quantile (Q-Q) plot [3] for the $z$-scores in Figure 6 and Figure 7 to demonstrate the distribution is indeed close to normal. The figures are shown in Appendix K.**  A Q-Q plot shows the quantiles of two distributions $X$ and $Y$ against each other, where in our case $X$ is normal, and $Y$ is the distribution of $z$-scores under an algorithm and a regularizer. If the plot approximately follows the line $y=x$, then the two distributions have approximately the same cumulative distribution function (CDF). In our case, this means that $Y$ is approximately normal. **Empirically, we observe that the Q-Q plot is very close to the line $y=x$, thus the distribution of $z$-scores is close to a normal distribution. As a result, the $t$-statistic we calculate from our samples is close to the $t$-distribution with parameter $n$, which converges to $\mathcal{N}(0,1)$ quickly as $n$ increases.**
> > >
> > > In Figure 6, we demonstrate that, for our results in Section 3 under the default hyperparameters, $\\{f_{\text{reg}}(e, s): e\in E, s\in S\\}$ and $\\{f_{\text{baseline}}(e, s): e\in E, s\in S\\}$ are close to normal distributions. In Figure 7, we demonstrate that, for our results in Section 4 over multiple sampled hyperparameters, $\\{f_{\text{reg}}(h, e, s): h\in H, e\in E, s\in S\\}$ and $\\{f_{\text{baseline}}(h, e, s): h\in H, e\in E, s\in S\\}$ are close to normal.
> > >
> > >
> > > [1] https://en.wikipedia.org/wiki/Welch%27s_t-test
> > > [2] https://en.wikipedia.org/wiki/Location_test
> > > [3] Q-Q plot. https://en.wikipedia.org/wiki/Q%E2%80%93Q_plot

---

> > > > ### Comment · AnonReviewer2 · 2020-11-22
> > > > **response to QQ-Plot**
> > > >
> > > > Thank you very much for the Q-Q plots! I appreciate all the work you've put in to addressing my concerns and those of the other reviewers. I feel that my concern regarding normality has now been addressed, which I will take into account when updating my score.

---

### Official Review · AnonReviewer1 · 2020-10-28
**Good empirical study**

**Rating:** 7
**Confidence:** 5

**Review:**

The paper studies how different regularizations affect common RL algorithms in continuous control tasks.

The paper is very well written and organized. The authors clearly state the scope of the paper and its placement with respect to RL literature.
Despite having no theoretical novelty, the paper has a good empirical contribution by being the first comprehensive study of the subject.
I appreciate the comments and explanations the authors give about the results, and I believe that the findings of this paper can help the RL community in being more aware of the importance of regularization, which is often overlooked. This is crucial especially when it comes to reimplementing existing algorithms without relying too much on handtuned hyperparameters.

Overall, I find this paper to be a good empirical study and I am leaning to accept it. However, my major concern is the number of seeds per experiment.
As shown by Henderson et al. ("Deep Reinforcement Learning that Matters"), 5 seeds are definitely not enough to get accurate statistical results out of RL experiments. I understand that the amount of total experiments is extremely large, given all the different hyperparameters you are testing, but you should have aimed for at least 10 seeds, especially because the paper is based on empirical contributions.

As a further suggestion, I invite you to include DDPG, maybe in the final version of the paper. First, because it is another off-policy algorithm (you have 3 on-policy and 1 off-policy). Second, because (as you also wrote) DDPG was one of the few algorithms to include BN, while TRPO, PPO and SAC rely on more sophisticated regularizations (entropy, KL, or surrogate). This could lead to very interesting results.

Another suggestion (for future work, as it may be a bit out of scope) is to test all algorithms on sparse-reward environments. In this case, it is known that common algorithms either prematurely convergence to local optima, or do not learn anything at all. Some of the regularizations (eg, entropy) should be beneficial in this case, but an extensive study is still missing.

** EDIT **
The authors have addressed my concernes and I have increased my score.

---

> ### Author Response · Authors · 2020-11-20
> **Thanks for your review; addressing concerns below**
>
> We sincerely thank you for your constructive comments. We are encouraged that you found our motivation convincing, our experiments substantive, and our paper well-written. We would like to address the comments and questions below, and we have updated our submission accordingly.
>
> >1. My major concern is the number of seeds per experiment. As shown by Henderson et al. ("Deep Reinforcement Learning that Matters"), 5 seeds are definitely not enough to get accurate statistical results out of RL experiments ... you should have aimed for at least 10 seeds
>
> **In our updated version, we increase the number of seeds from 5 to 10** and present results for the six MuJoCo environments in Appendix G. We report $z$-scores, and we also conduct Welch’s $t$-test. We find that our observations are consistent with those in Section 3. For example, $L_2$ tops the average $z$-score most often, and by large margin in total; entropy regularization is best used with A2C; Dropout and BN are only useful in the off-policy SAC algorithm; the improvement over baseline is larger on hard tasks.
>
> **In our updated version, we also provided rigorous justification that, because we test on the entire set of environments instead of on a single environment, our sample size is large enough to satisfy the condition of Welch's $t$-test and provide reliable results.** We would like to refer to Appendix K and our response (1) to R2 for more details.
>
> >2. As a further suggestion, I invite you to include DDPG, maybe in the final version of the paper.
>
> **In our updated version, we have conducted experiments on DDPG [1], another off-policy algorithm, on MuJoCo. We report the results in Appendix M.** For baseline experiments, We take away the regularizations originally implemented in DDPG. We obtain similar observations as we did in SAC. Notably, Dropout and Batch Normalization can be useful in DDPG, as indicated by the higher average $z$-score than the baseline, which supports our hypothesis that they can be helpful on off-policy algorithms. Also, while the original DDPG paper uses BN, we find that Dropout is even more effective than BN.
>
> >3. Another suggestion (for future work, as it may be a bit out of scope) is to test all algorithms on sparse-reward environments.
>
> We thank the reviewer for pointing out this interesting research direction, and we look forward to pursuing this goal in future work. For example, experimenting our studied algorithms on the SparseAnt and SparseHumanoid environments.
>
> ---
>
> We hope our response could address your concern, and we thank you again for the helpful comments. We are glad to discuss further comments and suggestions.

---

### Official Review · AnonReviewer4 · 2020-10-28
**good experimental investigation but lack of insights**

**Rating:** 6
**Confidence:** 4

**Review:**

This work empirically studies the widely used regularization techniques for training deep neural networks, such as $L_2$/$L_1$ regularizer, Batch Normalization (BN), Weight Clip, and Dropout, in policy optimization algorithms (A2C, SAC, TRPO, PPO). The experimental results demonstrate that these Deep Learning (DL) regularizations actually can help policy optimization.

Pros:
1. The combination of DL regularizations and Reinforcement Learning (RL) algorithms seems to be a reasonable and under-explored idea. The motivation is convincing.
2. The authors conducted substantive experiments, which I appreciate.
3. The work is presented clearly, and the paper is well written.

Cons:
1. The explanations for why these DL regularizers work or not are hand-waving. As an empirical study paper, I understand theory is not the main focus. But since this paper focused on policy optimization, I expected some insights or explanations from RL perspectives, which are important to guide future research, but they are not provided here (or I was missing something).

    (1a) The DL regularizers studied in this paper have proved to help training neural networks. As neural networks are used as function approximations in RL, it is as expected sometimes they should have some improvements.

    (1b) The main reason the authors claimed for why some DL regularizers work is from the generalization perspective, which makes sense in DL. However, for policy optimization, more explanations are needed from the perspectives of learning better agents (e.g., exploration vs. exploitation, and better objective landscape), which make more sense in RL. An interpretation from an RL perspective is lacking in this paper, which seems necessary since policy optimization is the main topic of this paper.

2. Experimental results are not enough to provide useful conclusions. Since the main focus is on the empirical side, I would expect more on this part, but it seems some conclusions have been made in this paper without sufficient investigations.

    (2a) The comparison of DL regularizers with entropy regularization actually does not seem reasonable to me.

    First, the entropy regularization is provable to increase exploration (see [1] to the end) and help convergence in policy optimization (see [2,3]), which is not claimed to help generalization. Second, DL regularizers help generalization as claimed in the paper. Therefore, they help agent learning in different ways, and I did not see the reason to compare them and what we can conclude from the results.

    (2b) The conclusion that DL regularizers do not work very well for value function (comparing with policy optimization) is lack of support.

    There is a number of regularizers/tricks of training in value functions (e.g., replay buffer, multi-step roll-out, distributional RL, double-Q, etc, see [4]). The authors did not do experiments (or did not mention) using those well-known ideas in RL and made this conclusion, which seems hasty to me.

    (2c) The conclusion and explanation that BN does not work for on-policy methods and works better for off-policy methods seem quite interesting. But also the study here is not enough. There is an amount of RL techniques for off-policy training (e.g., corrections, see [5]). I would suggest more investigation and deeper explanation than the discussion of the paper in this direction.

Overall, the idea of using DL regularizers in RL seems reasonable and the experimental results look promising. However, the theory part is not solid and insightful, and some of the conclusions are lack support.

References:
    [1] "Making sense of reinforcement learning and probabilistic inference", O’Donoghue et al.
    [2] "Understanding the impact of entropy on policy optimization", Ahmed et al.
    [3] "On the global convergence rates of softmax policy gradient methods", Mei et al.
    [4] "Rainbow: Combining Improvements in Deep Reinforcement Learning", Hessel et al.
    [5] "Safe and Efficient Off-Policy Reinforcement Learning", Munos et al.


======Update======
Thank you for the rebuttal, which resolved most of my concerns. I increased my score.

---

> ### Author Response · Authors · 2020-11-20
> **Thank you for your review; addressing concerns below [2/2]**
>
> >2a. First, the entropy regularization is provable to increase exploration and help convergence in policy optimization, which is not claimed to help generalization. Second, DL regularizers help generalization as claimed in the paper ... Comparison of DL regularizers with entropy regularization actually does not seem reasonable.
>
> We would like to analyze why we believe comparing entropy regularization with DL regularizers is meaningful from the following perspectives:
>
> - **Entropy regularization can arguably be seen as a form of preventing overfitting / helping generalization.** According to [1], entropy regularization “improves exploration by discouraging premature convergence to suboptimal deterministic policies”. Therefore, entropy prevents the policy network from overfitting to certain deterministic actions given certain states, which arguably can be seen as a form of helping generalization.
> - **DL regularizers could encourage exploration while helping generalization.** In Table 5 and Figure 5, we show that $L_2$ regularization not only decreases policy norm, but also increases policy entropy. This suggests that DL regularizers could accomplish the effect of entropy regularization. Since entropy regularization encourages exploration, DL regularizers could also encourage exploration.
> - **We empirically observe that entropy regularization and some DL regularizers could have overlapping effects.** In Appendix N, we observe that using entropy regularization and $L_2$ regularization together is at most marginally more effective than using entropy regularization or $L_2$ regularization alone, so they might have similar effect throughout the learning process.
>
> >2b. The conclusion that DL regularizers do not work very well for value function (comparing with policy optimization) is lack of support. There is a number of regularizers/tricks of training in value functions (e.g., replay buffer, multi-step roll-out, distributional RL, double-Q, etc). The authors did not do experiments (or did not mention) using those well-known ideas in RL and made this conclusion.
>
> In our work, we investigate the standard version of A2C, TRPO, PPO, and SAC as described in the original papers and as implemented in OPENAI Baselines [2]. **Indeed, SAC adopts the replay buffer and double-Q. A2C, TRPO, and PPO adopt multi-step roll-out, and the sum of discounted rewards is used as the objective for training the value network.** In Table 4, we empirically observe that the value function regularization does not perform well on the four algorithms we study, even though they adopted tricks for training value functions. In the updated version, we have clarified this point in Section 5.
>
> Despite that some of these tricks are used in the algorithms we study, analyzing the individual effect of these tricks in training the value function is not the main focus of our current work. We would like to leave the interaction between these individual techniques and value network regularization for future work.
>
> >2c. The conclusion and explanation that BN does not work for on-policy methods and works better for off-policy methods seem quite interesting. But also the study here is not enough.
>
> **In the updated version, we have added experiments on DDPG [4], another off-policy algorithm, on MuJoCo. We report the results in Appendix M.** DDPG uses the technique of "soft updates" to encourage training stability when updating the target Q network,  rather than directly copying the weights from the current Q network. We obtain similar observations as we did in SAC. Notably, Batch Normalization and Dropout can be useful in DDPG. BN can improve the average $z$-score by 0.54 over the baseline, and Dropout can improve the average $z$-score by 1.18. The results are statistically significant. This further supports our hypothesis that they can be helpful on off-policy algorithms.
>
> ---
>
> We hope our response could address your concern, and we thank you again for the helpful comments. We are glad to discuss further comments and suggestions.
>
> [1] Minh et al., Asynchronous Methods for Deep Reinforcement Learning
> [2] https://github.com/openai/baselines
> [3] Bellemare et al., A Distributional Perspective on Reinforcement Learning
> [4] Lillicrap et al., Continuous Control with Deep Reinforcement Learning

---

> ### Author Response · Authors · 2020-11-20
> **Thank you for your review; addressing concerns below [1/2]**
>
> We sincerely thank you for your constructive comments. We are encouraged that you found our motivation convincing, our experiments substantive, and our writing clear. We first would like to note that we provide some insights from the perspectives of sample complexity, return distribution, weight norm, policy entropy, and noise robustness in Section 6. We would like to address the comments and questions below, and we have updated our submission accordingly.
>
> >1. I expected some insights or explanations from RL perspectives
>
> **We believe that we have analyzed the effectiveness of these DL regularizers from several RL perspectives.** For example, we study how the effectiveness of DL regularizers differs between on-policy and off-policy algorithms, and between the policy and the value network. We also conduct analyses that are specific to RL. We study how DL regularizers affect the return distribution over multiple trajectories (Figure 4). We study how RL-specific data augmentation affects the performance of the baseline and the regularized models (Table 6). We also study how DL regularizers such as $L_2$ has an effect on the policy output entropy (Table 5, Figure 5), thus can lead to more policy exploration, similar to entropy regularization.
>
> >1a. The DL regularizers studied in this paper have proved to help training neural networks. As neural networks are used as function approximations in RL, it is as expected sometimes they should have some improvements.
>
> The regularizers we study in our paper (e.g. $L_2$ regularization, dropout) are the ones that have proved very effective in supervised learning settings. **Despite that supervised learning and RL both use neural networks, we believe that the RL setting still significantly differs from the supervised learning setting.** For example, the regression target of supervised learning is fixed for the same input, but the regression target for RL is always moving for the same state and action pair; in supervised learning, the network does not interact with the environment, while in RL, the agent constantly interacts with the environment. Therefore, due to a variety of differences, it is possible that regularizers effective in supervised learning are not very effective in RL (e.g. BN and dropout in on-policy algorithms). Previously, benchmark algorithms (A2C, PPO, TRPO, SAC) and libraries (OpenAI Baselines) did not consider using these regulariziers, which may be partly due to this reason.
>
> >1b. More explanations are needed from the perspectives of learning better agents (e.g., exploration vs. exploitation, and better objective landscape).
>
> We would like to point to an experiment that demonstrated $L_2$ can also lead to more exploration, in Table 5 and Figure 5. In our updated paper, we plot the policy entropy value for the baseline, $L_2$ regularization and entropy regularization throughout the training process. **Interestingly, we observe that $L_2$ regularization also has an effect on the policy output entropy, thus can lead to more policy exploration, similar to entropy regularization.** However, we still observe $L_2$ regularization to outperform entropy regularization, which could be due to the fact that it also leads to smaller weight norm, which may prevent overfitting.

---

### Official Review · AnonReviewer3 · 2020-10-31
**Good benchmark paper that comprehensively evaluates the impact of regularization**

**Rating:** 7
**Confidence:** 3

**Review:**

This paper conducts a comprehensive study on the effect of different regularization on Deep RL algorithms. Regularization has been mostly neglected in RL as most benefits were believed to be in generalization to unseen test environments in supervised learning settings. However, this paper shows that regularization does provide benefit even though training/testing is done on the same environment in deep RL settings.
The paper studies L1/L2 regularization, dropout, weight clipping, and batch normalization on four different deep RL policy optimization algorithms. Results show that regularization improves performance especially L2, that regularization brings more benefit on harder tasks that have higher sample complexity, and that it makes algorithms more robust to training hyperparameter variations. The paper conducts rigorous statistical significance test, and analyzes the benefit of regularization through four ways: sample complexity, reward distribution, weight norm, and training noise robustness.

Overall, I think the paper is well-written giving a comprehensive evaluation on a widely neglected area in reinforcement learning. Many different RL algorithm implementations have used regularization with and without acknowledging its use in the paper, and this paper sheds light that using regularization in deep RL algorithms does have significant impact and warrants further study.

I wish the paper conducted more than 5 runs (it is shown that mujoco environments have high variance due to random seeds and that same alg. performs significantly differently based on different groups of random seeds -- see Henderson et al. 2018 (https://arxiv.org/pdf/1709.06560.pdf)), but the authors also perform significance testing to validate the performance improvements.

I think the paper's analysis section can be further improved. For example, the bar chart in Figure 3 does not indicate how many runs it has done, or show any error bars to show statistical significance. And I think 'return' would be the correct terminology instead of 'reward' to indicate cumulative sum of rewards (For Figure 2, 3, and Table 5)

Also, I think the claim that 'BN and dropout work only with off-policy algorithm', or 'BN and dropout can only help in off-policy algorithm' is quite strong. Only a single off-policy algorithm SAC has been tested, and although BN and dropout helped a lot, its improvement may have been due to algorithm-specific properties of SAC. The authors hypothesize plausible reasons, but I think the findings only show that BN and dropout does not work well for on-policy algorithms.

There is a minor typo in page 4: decompled -> decoupled

---

> ### Author Response · Authors · 2020-11-20
> **Thanks for your review; addressing concerns below**
>
> We sincerely thank you for your constructive comments. We are encouraged that you found our statistical significance test rigorous, our evaluation comprehensive, and our paper well-written. We would like to address the comments and questions below, and we have updated our submission accordingly.
>
> >1. I wish the paper conducted more than 5 runs (it is shown that mujoco environments have high variance due to random seeds and that same alg. performs significantly differently based on different groups of random seeds -- see Henderson et al. 2018.
>
> **In our updated version, we increase the number of seeds from 5 to 10** and present results for the six MuJoCo environments in Appendix G to further verify our observations. We report $z$-scores, and we also conduct Welch’s $t$-test. We find that our observations in Section 3 still hold. For example, $L_2$ tops the average $z$-score most often, and by large margin in total; entropy regularization is best used with A2C; Dropout and BN are only useful in the off-policy SAC algorithm; the improvement over baseline is larger on hard tasks.
>
> In our updated version, we also provided rigorous justification that, because we test on the entire set of environments instead of on a single environment, our sample size is large enough to satisfy the condition of Welch's $t$-test and provide reliable results. We would like to refer to Appendix K and our response (1) to R2 for more details. In addition, we address that the unique environment property of HalfCheetah contributes to Henderson et al.'s observation that the same algorithm performs significantly using different sets of random seeds. We would like to refer to Appendix K and our response (3b) to R2 for more details.
>
> >2. The bar chart in Figure 3 does not indicate how many runs it has done, or show any error bars to show statistical significance.
>
> **We have updated Figure 3 to include error bars from our increased 10 random seeds.** For some environments, we find that adding proper regularization can significantly reduce the error bars (SAC Ant, PPO Humanoid).
>
> >3. 'Return' would be the correct terminology instead of 'reward' to indicate cumulative sum of rewards.
>
> We have revised the expressions accordingly.
>
> >4. Only a single off-policy algorithm SAC has been tested, and although BN and dropout helped a lot, its improvement may have been due to algorithm-specific properties of SAC.
>
> **In the updated version, we have added experiments on DDPG [1], another off-policy algorithm, on MuJoCo. We report the results in Appendix M.** We obtain similar observations as we did in SAC. Notably, Dropout and Batch Normalization can be useful in DDPG, as indicated by the higher average $z$-score than the baseline, which supports our hypothesis that they can be helpful on off-policy algorithms.
>
> >5. Typo on Page 4
>
> We have corrected the typo accordingly.
>
> ---
>
> We hope our response could address your concern, and we thank you again for the helpful comments. We are glad to discuss further comments and suggestions.
>
> [1] Lillicrap et al., Continuous Control with Deep Reinforcement Learning

---

> > ### Comment · AnonReviewer3 · 2020-11-24
> > **Thanks for the update**
> >
> > Thank you for addressing all my concerns and other reviewers'!
> > The paper looks much better.

---

### Author Response · Authors · 2020-11-25
**Summary of response & revision**

Thanks to all reviewers for their constructive comments! We have updated our submission accordingly. We would like to give a summary of main questions and address the main concerns:

>1. The number of seeds for our experiments might not be enough (R1, R2, R3)

In our updated version, we increase the number of seeds from 5 to 10 and present results for the six MuJoCo environments in Appendix G. We report $z$-scores, and we also conduct Welch’s $t$-test. We find that our observations in Section 3 still hold, i.e. our observations from 5 seeds per environment are consistent with those from 10 seeds per environment.

>2. The statistical significance of our results might not be reliable (R1, R2)

In our updated version, we also provided theoretical and empirical justification that, because we test on the entire set of environments instead of on a single environment, our sample size is large enough to satisfy the condition of Welch's $t$-test and provide reliable results. We would like to refer to Appendix K and our response (1) to R2 for more details.


>3. Request for DDPG experiments (R1); the study that BN and dropout work better for off-policy algorithm is not enough (R3, R4)

In the updated version, we have added the experiments on DDPG, an off policy algorithm, and reported the results in Appendix M. We obtain similar observations as we did in SAC. Notably, Batch Normalization and Dropout can be useful in DDPG. This further supports our hypothesis that they can be helpful on off-policy algorithms.

>4. R4’s concerns on lack of insight from RL perspectives; comparison with entropy regularization not reasonable; lack of value network training tricks.

We clarify our study and analysis from RL perspectives: policy/value regularization (Section 5), on/off policy, return distribution, RL data augmentation (Section 6). We further address this by showing $L_2$ also increases the policy entropy and encourages exploration (Table 5, Figure 5). This also partly justifies the comparison with entropy regularization. We further justify the comparison by pointing out that $L_2$ and entropy regularization can have overlapping effect (Appendix N). We clarify that our evaluated algorithms indeed adopt some of the tricks for training value networks (Section 5).

>5. RL results can have high variance according to Henderson et al. (R1, R2, R3)

Besides increasing our seeds from 5 to 10, we also address that the unique environment property of HalfCheetah contributes to Henderson et al.'s observation that the same algorithm performs significantly using different sets of random seeds. We would like to refer to Appendix K and our response (3b) to R2 for more details. Empirically, we found that other environments do not yield as much fluctuations as HalfCheetah.

---

### Decision · Program_Chairs · 2021-01-07
**Final Decision**

**Decision:**

Accept (Spotlight)

**Comment:**

The reviewers all  appreciated the insights drawn from this study as well at its thoroughness. I want to commend both authors for running additional experiments to strengthen the paper and reviewers for updating the scores accordingly.

Congratulations.